# Paradigm Shift of GNN Explainer from Label Space to Prototypical Representation Space

**Jun Yin**[*1,2], **Senzhang Wang**[†1], **Ziluowen Luo**[1], **Peng Huo**[4], **Hao Yan**[1], **Hao Miao**[2],
**Chaozhuo Li**[5], **Shirui Pan**[3], **Chengqi Zhang**[2]
[1]Central South University, [2]Hong Kong Polytechnic University, [3]Griffith University,
[4]National Super Computing Center, [5]Beijing University of Posts and Telecommunications
`yinjun2000@csu.edu.cn,szwang@csu.edu.cn,lzlwddl@csu.edu.cn,`
`huopeng@nscc-tj.cn,csuyh1999@csu.edu.cn,hao.miao@polyu.edu.hk,`
`lichaozhuo@bupt.edu.cn,s.pan@griffith.edu.au,chengqi.zhang@polyu.edu.hk`

## Abstract

Post-hoc instance-level graph neural network (GNN) explainers are developed to identify a compact subgraph (i.e., explanation) that encompasses the most influential components for each input graph. A fundamental limitation of existing methods lies in the insufficient utilization of structural information during GNN explainer optimization. They typically optimize the explainer by aligning the GNN predictions of input graph and its explanation in the graph label space which inherently lacks expressiveness to describe various graph structures. Motivated by the powerful structural expression ability of vectorized graph representations, we for the first time propose to shift the GNN explainer optimization from the graph label space to the graph representation space. However, the paradigm shift is challenging due to both the entanglement between the explanatory and non-explanatory substructures, and the distributional discrepancy between the input graph and the explanation subgraph. To this end, we meticulously design **IDEA**[1], a universal dual-stage optimization framework grounded in a prototypical graph representation space, which can generalize across diverse existing GNN explainer architectures. Specifically, in the Structural Information Disentanglement stage, a graph tokenizer equipped with a structure-aware disentanglement objective is designed to disentangle the explanatory substructures and encapsulate them into explanatory prototypes. In the Explanatory Prototype Alignment stage, IDEA aligns the representational distributions of the input graph and its explanation unified in the prototypical representation space, to optimize the GNN explainer. Comprehensive experiments on real-world and synthetic datasets demonstrate the effectiveness of IDEA, with the average improvements of ROC-AUC by 4.45% and precision by 48.71%. We further integrate IDEA with diverse explainer architectures and achieve an improvement by up to 10.70%, which verifies its generalizability.

## 1 Introduction

Post-hoc instance-level graph neural network (GNN) explainer (Ying et al., 2019; Luo et al., 2020; Schlichtkrull et al., 2021; Wang et al., 2021; Chen et al., 2023; Wang et al., 2023b; Zhang et al., 2023; Zhao et al., 2023; Chen et al., 2024) is a prominent research line to reveal the opaque decision-making mechanism of GNNs utilized in different domains (Fan et al., 2019; He et al., 2020b; Wu et al., 2023b; Liu et al., 2021; Yang et al., 2024b; Li et al., 2020; Mao et al., 2020). For each input graph, post-hoc instance-level GNN explainer aims to identify a compact explanation subgraph that is the most influential to the prediction made by the target GNN model.

Most existing GNN explainers are developed under the *label preserving framework* (Zhao et al., 2023; Zhang et al., 2023) as illustrated in Figure 1(a). Within this framework, a variety of explainer

---

[*]This research is initiated at *Central South University* and completed at *Hong Kong Polytechnic University*.

[†]Corresponding Author.
[1]Our code and datasets are available at https://github.com/Esperanto-mega/IDEA

architectures have been proposed. For example, GNNExplainer (Ying et al., 2019) determines the importance of edges and nodes through optimizable soft masks. PGExplainer (Luo et al., 2020) introduces a parametric graph generator to capture global explanatory structures. D4Explainer (Chen et al., 2023) combines the explanation search process with the denoising diffusion model (Ho et al., 2020). V-InFoR (Wang et al., 2023b) and ProxyExplainer (Chen et al., 2024) incorporate the variational graph auto-encoder (Kipf & Welling, 2016) to improve the robustness of GNN explainer.

Despite promising achievements, the label preserving framework exhibits a fundamental limitation in utilizing structural information to identify the explanation subgraphs, thus restricting the performance of GNN explainers. As shown in Figure 1(a), the label preserving framework optimizes the explainer by aligning the GNN predictions of the input graph and the explanation subgraph in the graph label space. Nevertheless, the graph label inherently lacks expressiveness to capture the characteristics of topological structures (Yang et al., 2024a; Wang et al., 2023a). During the GNN explanation process, the topological structures are critical, especially for complex graph domains such as molecular property prediction (Kazius et al., 2005; Agarwal et al., 2023; Wu et al., 2023a; Funke et al., 2023), where multiple distinct substructures can correspond to the same label.

In order to mitigate the limitation of label preserving framework, we advocate, for the first time, to shift the GNN explainer optimization framework from the graph label space to the graph representation space. Compared with discrete graph labels, the continuous graph representations can provide fine-grained descriptions of topological structures (Sun et al., 2020; Thakoor et al., 2022; Tian et al., 2022; Yang et al., 2024a). Consequently, developing a graph representation space based optimization framework is promising to facilitate the GNN explainer to sufficiently utilize structural information during explanation process. As shown in Figure 1(b), a straightforward implementa-

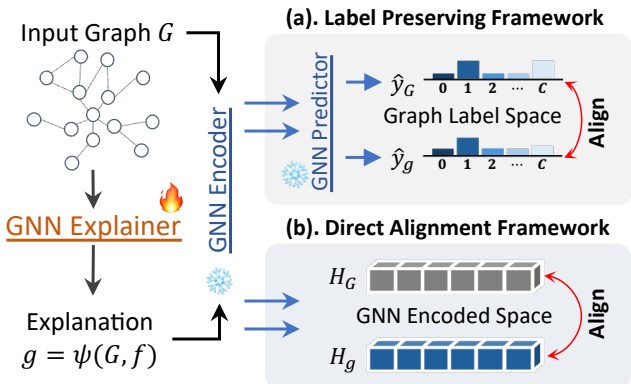

Figure 1: Overview of (a). Currently prevalent label preserving framework and (b). Direct alignment framework in GNN encoded representation space.

tion of this blueprint is the *direct alignment framework*, which optimizes the explainer by aligning the GNN encoded representations of the input graph and the corresponding explanation. However, the direct alignment framework is far from being an effective optimization framework for GNN explainers, due to the following two critical challenges.

The first challenge lies in the entanglement between the explanatory and non-explanatory substructures of the input graph. As revealed by causal inference theory (Wu et al., 2022; Sui et al., 2022), the explanatory substructure causally determines the GNN prediction, while the non-explanatory counterpart merely exhibits statistical correlations. Due to the message passing mechanism (Kipf & Welling, 2017; Veličković et al., 2018; Xu et al., 2019), the GNN encoded representation of the input graph inevitably aggregates explanatory and non-explanatory substructures. Directly aligning the representations of the input graph and the explanation risks misleading the GNN explainer to non-explanatory substructures. The second challenge arises from the distributional discrepancy between the input graph and its explanation subgraph within the GNN encoded representation space. Since the explanation subgraph is a structurally reduced version of the input graph, the explanation representation naturally follows a deviated distribution in the GNN encoded representation space (Zhang et al., 2023; Chen et al., 2024). Simplistically enforcing the representation similarity within the GNN encoded space tends to obscure the most influential subgraph rather than reveal it.

To overcome the challenges above, we propose **IDEA**, a universal dual-stage GNN explainer optimization framework grounded in a prototypical graph representation space, which is generalizable across various existing GNN explainer architectures. Specifically, IDEA consists of a Structural Information Disentanglement stage and an Explanatory Prototype Alignment stage. In the structural information disentanglement stage, we design a hierarchical graph tokenizer equipped with a customized structure-aware disentanglement objective, to disentangle the explanatory substructures

from confounding non-explanatory counterpart and then cluster them into prototypical representations. In the explanatory prototype alignment stage, IDEA first unifies the GNN encoded representations of the input graph and the explanation in the prototypical representation space, to mitigate the distributional discrepancy. Subsequently, IDEA aligns the unified representational distributions to optimize the GNN explainer, enabling accurate identification of GNN explanations.

The main contributions of this work are summarized as follows.

- We propose, for the first time, the paradigm shift of GNN explainer optimization framework from the graph label space to the graph representation space. Furthermore, we design IDEA, the first graph representation space based GNN explainer optimization framework.
- We propose a hierarchical graph tokenizer equipped with a structure-aware disentanglement objective, to disentangle the explanatory substructures and encapsulate them into prototypical representations. We formulate a novel explanation identification strategy based on the prototypical representation space, which aligns the unified representational distributions of the input graph and the explanation, to circumvent the deviated distribution of the explanation subgraph.
- Extensive experiments conducted on real-world and synthetic datasets validate the effectiveness of IDEA compared with SOTA GNN explainers, with the average improvements of ROC-AUC by 4.45% and precision by 48.71%. Meanwhile, the consistent superiority of the collaboration between IDEA and various explainer architectures demonstrates the generalizability of IDEA.

## 2 NOTATION AND PROBLEM FORMULATION

**Notation.** We use $G = (A, X)$ with the adjacency matrix $A \in \mathbb{R}^{N \times N}$ and the feature matrix $X \in \mathbb{R}^{N \times D}$ to denote a graph data of $N$ nodes, where $D$ represents the graph feature dimension. If node $v_i$ and node $v_j$ are connected, the element in the $i$-th row and the $j$-th column $A_{ij} = 1$, and $0$ otherwise. Without losing generality, in this work, we focus on the *graph classification* task (Hu et al., 2022; Li et al., 2017), since node classification can be converted into a computation graph classification problem (Chen et al., 2024). For graph classification, each graph $G$ is associated with a label $y \in \mathbb{R}^{1 \times C}$ where $C$ denotes the total number of classes. The target graph neural network model $f(\cdot)$ has been well-trained to predict the class of any given graph $G$. Generally, the to-be-explained GNN model consists of the following three modules, the feature encoder $f_e(\cdot)$, the pooling function $\text{Pool}(\cdot)$ (e.g., mean pooling and max pooling) (Ying et al., 2018; Du et al., 2021), and the task predictor $f_p(\cdot)$. The GNN prediction procedure can be represented as follows,

$$H_N = f_e(G), H_G = \text{Pool}(H_N), \hat{y} = f_p(H_G), \tag{1}$$

where $H_N \in \mathbb{R}^{N \times d}$ is the matrix of $d$-dimensional node representations, $H_G \in \mathbb{R}^{1 \times d}$ is the pooled graph representation, and $\hat{y}$ is the predicted label. Refer to Appendix A for notation summary.

**Problem Formulation.** Given a well-trained GNN model $f(\cdot)$ to be explained and an input graph $G$, the post-hoc instance-level GNN explainer $\psi(\cdot, \cdot)$ aims to identify a compact subgraph $g^* = \psi(G, f) \subset G$, which retains the most influential components during the GNN predicting procedure. Within the label preserving framework, the identified subgraph is reinforced to maintain the original prediction of $G$. Typically, the optimization objective of the label preserving framework is defined as the mutual information between the predictions of the input graph and the explanation subgraph, i.e., $\text{MI}(f(g), f(G))$. In this work, we shift the GNN explainer paradigm from the label space to the graph representation space, to sufficiently utilize the structural information for GNN explanations.

## 3 METHODOLOGY

Procedurally, **IDEA** consists of two successive stages, the Structural Information Disentanglement and the Explanatory Prototype Alignment, centered on the hierarchical graph tokenizer (HGTokenizer). To tackle the structural entanglement problem, in the first stage, we design a structure-aware disentanglement (SAD) objective for HGTokenizer to stratify the explanatory and non-explanatory substructures. During the disentanglement process, the explanatory substructures are clustered into a collection of explanatory prototypes. In the second stage, based on the HGTokenizer and the prototypes, we first unify the representations of the input graph and the explanation subgraphs into the prototypical representation space, to circumvent the distribution discrepancy problem. Afterwards, the GNN explainer is optimized by aligning the two unified representational distributions.

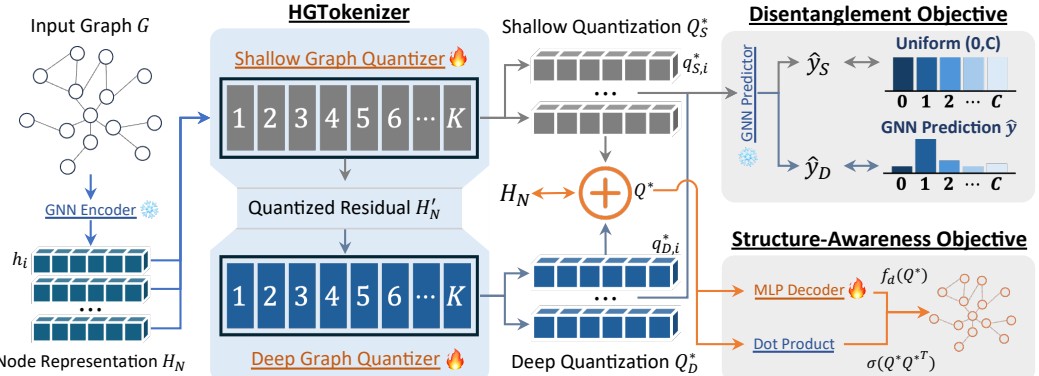

Figure 2: Overview of the Structural Information Disentanglement in IDEA. The node representation $H_N$ is decomposed into two quantization representations $Q_S^*$ and $Q_D^*$, by the cascade-connected graph quantizers in HGTokenizer. The two quantizers aim to capture the explanatory and non-explanatory substructures respectively, following the guidance of the SAD objective.

## 3.1 Structural Information Disentanglement

In Figure 2, we outline the overview of structural information disentanglement, which empowers the HGTokenizer with the ability to decouple the explanatory substructures from the non-explanatory counterpart. Technically, the HGTokenizer is consist of two cascade-connected graph quantizers (Zeghidour et al., 2021) which take insights of the semantic tokenization (Rajput et al., 2023; Yin et al., 2025), to compactly represent the structural information with discrete codebooks.

Given the node representation matrix $H_N$, HGTokenizer approximates it based on the shallow and the deep graph quantizers. For the representation to be quantized, the graph quantizer looks up the nearest codeword in the codebook. Since the codebook scale $K$ is significantly smaller than the total number of nodes, it can serve as a collection of prototypes (Dai & Wang, 2025; Zhu et al., 2025) that succinctly summarizes the input representations. Using the representation $h_i$ of node $v_i$ as an example, the cascade quantization procedure of HGTokenizer is formulated as follows,

$$q_{S,i}^* = \mathrm{GQ_S}(h_i) = \underset{q \in \mathcal{C}_S}{\arg\min} \, \mathcal{D}(h_i, q), \; h_i' = h_i - q_{S,i}^* \tag{2}$$

$$q_{D,i}^* = \mathrm{GQ_D}(h_i') = \underset{q \in \mathcal{C}_D}{\arg\min} \, \mathcal{D}(h_i', q), \; q_i^* = q_{S,i}^* + q_{D,i}^*, \tag{3}$$

where $\mathrm{GQ_S}(\cdot)$ and $\mathrm{GQ_D}(\cdot)$ denote the shallow graph quantizer and the deep graph quantizer, $\mathcal{C}_S$ and $\mathcal{C}_D$ denote the codebooks of quantizers, $q$ denotes the codeword inside, and $\mathcal{D}(\cdot, \cdot)$ is the distance metric for quantization. The deep graph quantizer takes the residual of the shallow one, in order to spontaneously dichotomize the fused representations encoded by the target GNN model.

The SAD objective utilized to optimize the HGTokenizer consists of three terms, i.e., the structure-awareness objective, the disentanglement objective, and the standard quantization objective. The *structure-awareness objective* $\mathcal{L}_S$ aims to recover the topological structures and node features based on the quantized node representations, enhancing the ability of HGTokenizer to capture the graph structural characteristics. Formally, $\mathcal{L}_S$ is defined as follows,

$$\mathcal{L}_S = \left\| A - \sigma(Q^* Q^{*T}) \right\|_2^2 + \left\| X - f_d(Q^*) \right\|_2^2, \tag{4}$$

where $Q^*$ is the matrix of quantized node representations $q_i^*$, $\sigma(\cdot)$ is the sigmoid function, and $f_d(\cdot)$ is a linear decoder. The *disentanglement objective* $\mathcal{L}_D$ enforces the prediction of the non-explanatory substructures towards a uniform distribution, and guides the prediction of the explanatory substructures towards the original prediction. Formally, $\mathcal{L}_D$ is defined as follows,

$$\mathcal{L}_D = \mathrm{KL}\big[\hat{y}_S \| \mathcal{U}_C\big] + \mathrm{CrossEntropy}\big(\hat{y}_D, \hat{y}\big), \tag{5}$$

where $\hat{y}_S$ and $\hat{y}_D$ denote the GNN predictions of the shallow and deep quantized representations, respectively. $\mathcal{U}_C$ denotes the uniform distribution in the graph label space.

By minimizing the Kullback-Leibler divergence between $\hat{y}_S$ and $\mathcal{U}_c$, IDEA reinforces the shallow graph quantizer to capture non-explanatory substructures that are unable to determine the GNN decision-making process. Meanwhile, the second term instructs the deep graph quantizer to encapsulate the explanatory substructures that are more influential. Consequently, the codebook $\mathcal{C}_D$ inside $\text{GQ}_\text{D}$ can not only maintain the GNN prediction of the input graph, but also recover the graph topological structures along with $\mathcal{C}_S$. IDEA regards $\mathcal{C}_D$ as a collection of explanatory prototypes which naturally induces a prototypical representation space for the GNN explainer optimization.

In addition, following the standard vector quantization process (van den Oord et al., 2017; Zeghidour et al., 2021), the *quantization objective* $\mathcal{L}_Q$ below is adopted for the basic quantization ability,

$$\mathcal{L}_Q = \left\| H_N - Q^* \right\|_2^2. \tag{6}$$

Hence, the structure-aware disentanglement objective is defined as follows,

$$\mathcal{L}_\text{SAD} = \mathcal{L}_D + \lambda_S \cdot \mathcal{L}_S + \lambda_Q \cdot \mathcal{L}_Q, \tag{7}$$

where $\lambda_S, \lambda_Q$ are the weighted hyper-parameters. We present a hyper-parameter analysis on the weights $\lambda_S$ and $\lambda_Q$ of the structure-aware disentanglement objective in Appendix C.1.

## 3.2 EXPLANATORY PROTOTYPE ALIGNMENT

Following the guidance of the SAD objective, the HGTokenizer can disentangle the explanatory information from the fused graph representation encoded by the target GNN. The deep quantizer further encompasses a collection of prototypes to describe the explanatory information. To circumvent the deviated distribution of the explanation subgraphs, we formulate a novel explanation identification strategy on the basis of the prototypical representation space. The overview of the explanatory prototype alignment is illustrated in Figure 3.

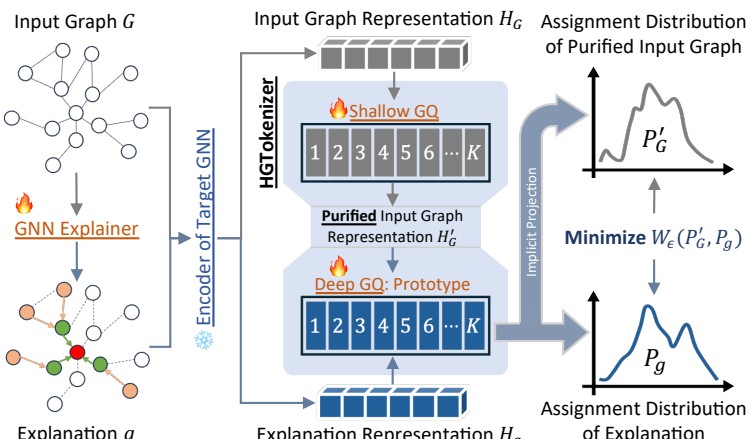

Figure 3: Overview of the Explanatory Prototype Alignment in IDEA. The input graph representation $H_G$ is first purified by the shallow graph quantizer, to eliminate the non-explanatory information. Then, the explanation representation $H_g$ and the purified input graph representation $H'_G$ are implicitly projected into the prototypical space. At last, IDEA aligns the assignment distributions $\mathcal{P}'_G$ and $\mathcal{P}_g$ to optimize the explainer.

Given the target GNN model $f(\cdot)$ to be explained and an input graph $G$, the explanation subgraph $g$ is generated by $\psi(G, f)$, where $\psi$ denotes the GNN explainer. In our implementation, a typical probabilistic generator, which is well-investigated among the GNN explanation community (Luo et al., 2020; Wang et al., 2021; 2023b; Wang & Shen, 2023), is adopted as the GNN explainer backbone. The implementation details are elaborated in Appendix B.3. Formally, we denote the GNN encoded representation of the input graph as $H_G$ and that of the explanation subgraph as $H_g$.

To filter out the non-explanatory information from the input graph representation, we feed $H_G$ to the HGTokenizer (i.e., the cascade of $\text{GQ}_\text{S}$ and $\text{GQ}_\text{D}$), formulated as follows,

$$H_{S,G} = \text{GQ}_\text{S}(H_G), \; H'_G = H_G - H_{S,G}, \; H_{D,G} = \text{GQ}_\text{D}(H'_G), \tag{8}$$

where $H_{S,G}$ is the non-explanatory fraction of the input graph representation and $H'_G$ is the purified input graph representation after removing $H_{S,G}$. For the explanation representation $H_g$, we directly feed it into the deep graph quantizer $\text{GQ}_\text{D}$, formulated as follows,

$$H_{D,g} = \text{GQ}_\text{D}(H_g). \tag{9}$$

Based on the quantization procedure of $GQ_D$, we can implicitly unify the purified representation of the input graph $H'_G$ and the explanation representation $H_g$ into the prototypical representation space, instead of explicit representation projection. To be more specific, the assignment distribution of the to-be-quantized representation (i.e., $H'_G$ or $H_g$) over the explanatory codebook $\mathcal{C}_D \in \mathbb{R}^{K \times d}$ is able to indicate its location within the prototypical representation space. Formally, the assignment distributions corresponding to $H'_G$ and $H_g$ are measured as follows,

$$\mathcal{P}'_G = \mathrm{Norm}\big(\mathcal{D}(H'_G, \mathcal{C}_D)\big), \ \mathcal{P}_g = \mathrm{Norm}\big(\mathcal{D}(H_g, \mathcal{C}_D)\big), \tag{10}$$

where $\mathcal{P}'_G$ and $\mathcal{P}_g$ denote the assignment distributions and $\mathrm{Norm}(\cdot)$ normalizes the quantization distance to the probability value. Theoretical justification for this practice is presented in Appendix G. Since the assignment distributions $\mathcal{P}'_G$ and $\mathcal{P}_g$ are identically measured over the implicit prototypical representation space, the distribution discrepancy of the explanation subgraph in the GNN encoded space is ingeniously circumvented.

Subsequently, IDEA adopts the entropy-regularized Wasserstein distance (Reshetova et al., 2024) between $\mathcal{P}'_G$ and $\mathcal{P}_g$ as the optimization objective of the GNN explainer $\psi$. The Wasserstein distance not only encourages the consistency between the two assignment probabilities $\mathcal{P}'_G$ and $\mathcal{P}_g$, but also is insensitive to the sparsity issue of probabilistic distributions. For the stability of the explainer optimization, IDEA adopts the symmetric variant defined as follow,

$$\mathcal{L}_{\mathrm{IDEA}} = W_\epsilon(\mathcal{P}'_G, \mathcal{P}_g) + \frac{1}{2}\Big(W_\epsilon(\mathcal{P}'_G, \mathcal{P}'_G) + W_\epsilon(\mathcal{P}_g, \mathcal{P}_g)\Big). \tag{11}$$

$$W_\epsilon(\mathcal{P}'_G, \mathcal{P}_g) = \min_{\gamma \in \Pi(\mathcal{P}'_G, \mathcal{P}_g)} \sum_{i,j} \gamma_{ij} S_{ij} + \epsilon \sum_{i,j} \gamma_{ij} \log \gamma_{ij}. \tag{12}$$

$\Pi(\mathcal{P}'_G, \mathcal{P}_g)$ denotes the transport polytope and $S$ denotes the cost matrix defined as follows,

$$\Pi(\mathcal{P}'_G, \mathcal{P}_g) = \big\{\Pi \in \mathbb{R}_+^{K \times K} | \Pi\mathbf{1} = \mathcal{P}'_G, \Pi^T \mathbf{1} = \mathcal{P}_g\big\}, \ S_{ij} = (\mathcal{P}'_{G,i} - \mathcal{P}_{g,j})^2. \tag{13}$$

We implement IDEA as a dual-stage framework in order to avoid the counteraction between the optimization terms within $\mathcal{L}_{\mathrm{SAD}}$ and $\mathcal{L}_{\mathrm{IDEA}}$. In Appendix F.1, we further investigate a variant of IDEA where the two stages are conducted jointly.

# 4 EXPERIMENT

To comprehensively validate the practicality of IDEA , we conduct extensive experiments which are designed to investigate the following research questions.

- **RQ1:** How effective is IDEA compared to the label preserving based SOTA baselines?
- **RQ2:** How generalizable is IDEA collaborated with different explainer architectures?
- **RQ3:** How does each component of IDEA influence the overall explanation performance?

Furthermore, we present the hyper-parameter analysis, the explanation visualization, and the time complexity analysis in Appendix C, D, and E, respectively.

## 4.1 EXPERIMENTAL SETUP

**Dataset.** We evaluate IDEA and baselines on both real-world and synthetic datasets. The evaluated real-world datasets include Mutagenicity (Kazius et al., 2005), Benzene (Sanchez-Lengeling et al., 2020), Fluoride-Carbonyl (Sanchez-Lengeling et al., 2020), and Alkane-Carbonyl (Sanchez-Lengeling et al., 2020). The synthetic datasets is BA-2Motifs (Luo et al., 2020).

**Baseline.** The baselines include SOTA post-hoc instance-level GNN explainers based on various techniques, i.e., GNNExplainer (Ying et al., 2019), PGExplainer (Luo et al., 2020), GraphMask (Schlichtkrull et al., 2021), ReFine (Wang et al., 2021), V-InFoR (Wang et al., 2023b), D4Explainer (Chen et al., 2023), MixupExplainer (Zhang et al., 2023), ProxyExplainer (Chen et al., 2024).

**Evaluation.** Following the standard experimental settings (Luo et al., 2020; Chen et al., 2024), we train a 3-layer Graph Convolutional Network (GCN) model (Kipf & Welling, 2017) on each

Table 1: Explanation performance (ROC-AUC ↑) of IDEA and eight SOTA post-hoc instance-level GNN explainers on five datasets, in the form of mean$_{\pm\text{std}}$. **Average** reports the mean result over all the evaluated datasets. *Improvement* is defined as $(\texttt{IDEA} - \texttt{Best-Baseline})/\texttt{Best-Baseline}$. The superscript * indicates the improvement is statistically significant with the $p$-value less than 0.01. **Bold** font and underline highlight the best and the runner-up performance, respectively.

| Model | Mutagenicity | Benzene | Alkane | Fluoride | BA-2Motifs | Average |
|---|---|---|---|---|---|---|
| GNNExplainer | $0.6155_{\pm0.0087}$ | $0.6863_{\pm0.0126}$ | $0.6884_{\pm0.0055}$ | $0.5399_{\pm0.0102}$ | $0.5619_{\pm0.0162}$ | $0.6184_{\pm0.0103}$ |
| PGExplainer | $0.7016_{\pm0.0201}$ | $0.8855_{\pm0.0023}$ | $0.7446_{\pm0.0086}$ | $0.8091_{\pm0.0209}$ | $0.8594_{\pm0.0072}$ | $0.8000_{\pm0.0115}$ |
| GraphMask | $0.6377_{\pm0.0083}$ | $0.5523_{\pm0.0062}$ | $0.6311_{\pm0.0139}$ | $0.5843_{\pm0.0028}$ | $0.6119_{\pm0.0035}$ | $0.6035_{\pm0.0068}$ |
| ReFine | $0.6833_{\pm0.0052}$ | $0.8720_{\pm0.0262}$ | $0.7293_{\pm0.0077}$ | $0.5600_{\pm0.0117}$ | $0.6115_{\pm0.0027}$ | $0.6912_{\pm0.0104}$ |
| V-InfoR | $0.6075_{\pm0.0149}$ | $0.6642_{\pm0.0112}$ | $0.6507_{\pm0.0162}$ | $0.6437_{\pm0.0169}$ | $0.7755_{\pm0.0243}$ | $0.6683_{\pm0.0156}$ |
| D4Explainer | $0.5467_{\pm0.0279}$ | $0.7239_{\pm0.0165}$ | $0.7736_{\pm0.0059}$ | $0.7484_{\pm0.0099}$ | $0.7478_{\pm0.0174}$ | $0.7081_{\pm0.0128}$ |
| MixupExplainer | $0.5428_{\pm0.0074}$ | $0.5399_{\pm0.0020}$ | $0.7385_{\pm0.0043}$ | $0.5400_{\pm0.0002}$ | $0.8355_{\pm0.0129}$ | $0.6393_{\pm0.0035}$ |
| ProxyExplainer | $0.6948_{\pm0.0035}$ | $0.8593_{\pm0.0127}$ | $0.9334_{\pm0.0033}$ | $0.8804_{\pm0.0126}$ | $0.8717_{\pm0.0028}$ | $0.8479_{\pm0.0068}$ |
| Direct-Align | $0.6567_{\pm0.0068}$ | $0.8809_{\pm0.0008}$ | $0.3562_{\pm0.0160}$ | $0.7988_{\pm0.0042}$ | $0.8653_{\pm0.0060}$ | $0.7116_{\pm0.0056}$ |
| IDEA | $\mathbf{0.7379}^*_{\pm0.0084}$ | $\mathbf{0.9138}^*_{\pm0.0002}$ | $\mathbf{0.9355}_{\pm0.0030}$ | $\mathbf{0.8868}_{\pm0.0018}$ | $\mathbf{0.9541}^*_{\pm0.0107}$ | $\mathbf{0.8856}^*_{\pm0.0047}$ |
| *Improvement* | 5.17% | 3.20% | 0.22% | 0.73% | 9.45% | 4.45% |

Table 2: Explanation performance (Precision ↑) of IDEA and SOTA baselines across five datasets.

| Model | Mutagenicity | Benzene | Alkane | Fluoride | BA-2Motifs | Average |
|---|---|---|---|---|---|---|
| GNNExplainer | $0.0736_{\pm0.0030}$ | $0.1901_{\pm0.0024}$ | $0.0104_{\pm0.0013}$ | $0.0652_{\pm0.0019}$ | $0.1373_{\pm0.0034}$ | $0.0953_{\pm0.0022}$ |
| PGExplainer | $0.1038_{\pm0.0067}$ | $0.4484_{\pm0.0041}$ | $0.0761_{\pm0.0077}$ | $0.3253_{\pm0.0176}$ | $0.6072_{\pm0.0016}$ | $0.3122_{\pm0.0072}$ |
| GraphMask | $0.0748_{\pm0.0070}$ | $0.1373_{\pm0.0075}$ | $0.0104_{\pm0.0082}$ | $0.0443_{\pm0.0029}$ | $0.2337_{\pm0.0043}$ | $0.1001_{\pm0.0036}$ |
| ReFine | $0.0833_{\pm0.0058}$ | $0.1951_{\pm0.0272}$ | $0.1304_{\pm0.0123}$ | $0.3027_{\pm0.0117}$ | $0.5054_{\pm0.0033}$ | $0.2434_{\pm0.0119}$ |
| V-InFoR | $0.1230_{\pm0.0075}$ | $0.3195_{\pm0.0134}$ | $0.1304_{\pm0.0010}$ | $0.2374_{\pm0.0019}$ | $0.1380_{\pm0.0161}$ | $0.1897_{\pm0.0075}$ |
| D4Explainer | $0.2087_{\pm0.0299}$ | $0.3538_{\pm0.0107}$ | $0.0109_{\pm0.0061}$ | $0.3685_{\pm0.0003}$ | $0.3153_{\pm0.0173}$ | $0.2514_{\pm0.0106}$ |
| MixupExplainer | $0.0682_{\pm0.0083}$ | $0.1385_{\pm0.0018}$ | $0.0652_{\pm0.0038}$ | $0.2929_{\pm0.0034}$ | $0.3194_{\pm0.0105}$ | $0.1768_{\pm0.0034}$ |
| ProxyExplainer | $0.3365_{\pm0.0058}$ | $0.5908_{\pm0.0135}$ | $0.3261_{\pm0.0035}$ | $0.1486_{\pm0.0032}$ | $0.6229_{\pm0.0089}$ | $0.4050_{\pm0.0067}$ |
| Direct-Align | $0.0805_{\pm0.0050}$ | $0.5443_{\pm0.0009}$ | $0.0109_{\pm0.0057}$ | $0.4890_{\pm0.0028}$ | $0.5872_{\pm0.0054}$ | $0.3424_{\pm0.0025}$ |
| IDEA | $\mathbf{0.4020}^*_{\pm0.0063}$ | $\mathbf{0.7523}^*_{\pm0.0003}$ | $\mathbf{0.4565}^*_{\pm0.0161}$ | $\mathbf{0.6119}^*_{\pm0.0183}$ | $\mathbf{0.7885}^*_{\pm0.0201}$ | $\mathbf{0.6022}^*_{\pm0.0119}$ |
| *Improvement* | 19.47% | 27.34% | 39.99% | 25.13% | 26.59% | 48.71% |

dataset, as the target model to be explained. To evaluate the explanation quality, we reformulate the explanation task as an edge binary classification task. Edges that belong to the expert-notated ground truth are labeled as positive, and negative otherwise. Hence, we adopt the ROC-AUC score as the main metric to evaluate the explanation performance (Ying et al., 2019; Luo et al., 2020). Refer to Appendix B for the experimental details.

## 4.2 EXPLANATION PERFORMANCE (RQ1)

The evaluation result of IDEA and SOTA post-hoc instance-level GNN explainers is presented in Table 1, in terms of the ROC-AUC score. *Direct-Align* corresponds to the direct alignment framework in Figure 1(b), which optimizes the GNN explainer by directly aligning the GNN encoded representations of the input graph and the explanation.

The result sufficiently demonstrates the effectiveness of IDEA, which can consistently achieve the supreme performance on all the evaluated datasets. On average, the improvement of IDEA over the best baseline is 4.45%. For the Mutagenicity dataset, which is a complex molecular property prediction dataset, IDEA advances the explanation quality by up to 5.17%, compared to the benchmark-leading baseline PGExplainer (Luo et al., 2020). Despite the primitive explanation identification strategy, *Direct-Align* achieves the second-tier performance among the evaluated explainers, showcasing the considerable potential of GNN explainer optimization framework based on the graph representation space. On the other hand, the inferiority of *Direct-Align* to the top-tier explainers, including PGExplainer, ProxyExplainer (Chen et al., 2024), and IDEA, justify the necessity of further advance on the direct alignment framework.

In light of the critical importance of precision in the GNN explanation evaluation, we further report the result of IDEA and SOTA baselines in Table 2. In general, the average precision of IDEA is 0.6022, achieving a significant improvement by 48.69% over the runner-up ProxyExplainer. Specifically, for the Alkane-Carbonyl dataset, whose ground-truth explanation is the union of an alkane

Table 3: Explanation performance (ROC-AUC ↑) of IDEA with different explainer architectures.

| Model | Mutagenicity | Benzene | Alkane | Fluoride | BA-2Motifs | Average |
|---|---|---|---|---|---|---|
| PGExplainer | $0.7016_{\pm0.0201}$ | $0.8855_{\pm0.0023}$ | $0.7446_{\pm0.0086}$ | $0.8091_{\pm0.0209}$ | $0.8594_{\pm0.0072}$ | $0.8000_{\pm0.0115}$ |
| +IDEA | $\mathbf{0.7379}^{*}{}_{\pm0.0084}$ | $\mathbf{0.9138}^{*}{}_{\pm0.0002}$ | $\mathbf{0.9355}^{*}{}_{\pm0.0030}$ | $\mathbf{0.8868}^{*}{}_{\pm0.0018}$ | $\mathbf{0.9541}^{*}{}_{\pm0.0107}$ | $\mathbf{0.8856}^{*}{}_{\pm0.0047}$ |
| *Improvement* | 5.17% | 3.20% | 25.64% | 9.60% | 11.02% | 10.70% |
| ReFine | $0.6833_{\pm0.0052}$ | $0.8720_{\pm0.0262}$ | $0.7293_{\pm0.0077}$ | $0.5600_{\pm0.0117}$ | $0.6115_{\pm0.0027}$ | $0.6912_{\pm0.0104}$ |
| +IDEA | $\mathbf{0.7832}^{*}{}_{\pm0.0028}$ | $\mathbf{0.8759}^{*}{}_{\pm0.0197}$ | $\mathbf{0.8428}^{*}{}_{\pm0.0018}$ | $\mathbf{0.5809}^{*}{}_{\pm0.0094}$ | $\mathbf{0.6861}^{*}{}_{\pm0.0016}$ | $\mathbf{0.7538}^{*}{}_{\pm0.0067}$ |
| *Improvement* | 14.62% | 0.45% | 15.56% | 3.73% | 12.20% | 9.05% |
| V-InfoR | $\mathbf{0.6075}_{\pm0.0149}$ | $0.6642_{\pm0.0112}$ | $0.6507_{\pm0.0162}$ | $0.6437_{\pm0.0169}$ | $0.7755_{\pm0.0243}$ | $0.6683_{\pm0.0156}$ |
| +IDEA | $0.5734_{\pm0.0057}$ | $\mathbf{0.6713}_{\pm0.0103}$ | $\mathbf{0.6776}^{*}{}_{\pm0.0008}$ | $\mathbf{0.6483}_{\pm0.0111}$ | $\mathbf{0.7772}_{\pm0.0058}$ | $\mathbf{0.6696}_{\pm0.0059}$ |
| *Improvement* | -5.61% | 1.07% | 4.13% | 0.71% | 0.22% | 1.38% |
| ProxyExplainer | $0.6948_{\pm0.0035}$ | $0.8593_{\pm0.0127}$ | $0.9334_{\pm0.0033}$ | $0.8804_{\pm0.0126}$ | $0.8717_{\pm0.0028}$ | $0.8479_{\pm0.0068}$ |
| +IDEA | $\mathbf{0.7215}^{*}{}_{\pm0.0134}$ | $\mathbf{0.8864}^{*}{}_{\pm0.0099}$ | $\mathbf{0.9509}^{*}{}_{\pm0.0099}$ | $\mathbf{0.8931}^{*}{}_{\pm0.0104}$ | $\mathbf{0.8930}^{*}{}_{\pm0.0047}$ | $\mathbf{0.8690}^{*}{}_{\pm0.0081}$ |
| *Improvement* | 3.84% | 3.15% | 1.87% | 1.44% | 2.44% | 2.48% |

chain ($C_nH_{2n+2}$) and a carbonyl group (C=O), the improvement of IDEA is the highest over the five evaluated datasets, by up to 39.99%. This advancement demonstrates the ability of IDEA to explain graphs from complex domains. Similarly, the naive contestant *Direct-Align* maintains the moderate position among the evaluated GNN explainers.

## 4.3 GENERALIZABILITY ACROSS EXPLAINER ARCHITECTURE (RQ2)

We scrutinize the generalizability of IDEA by integrating various leading GNN explainer architectures and the evaluation result in terms of ROC-AUC is presented in Table 3. In detail, PGExplainer (Luo et al., 2020) adopts a well-investigated subgraph generator based on the concrete distribution (Maddison et al., 2017). ReFine (Wang et al., 2021) implements a subgraph generator for each graph class to capture the contrastive information. V-InFoR (Wang et al., 2023b) introduces a graph vairational auto-encoder (GVAE) to refine the GNN encoded representations for robustness to structural corruptions. ProxyExplainer (Chen et al., 2024) merges a GAVE and a standard graph auto-encoder as the proxy generator to resist distribution discrepancy caused by the explanation subgraph.

The evaluation result sufficiently demonstrates that IDEA is generalizable across four different GNN explainer architectures, with the average improvement by 10.70%, 9.05%, 1.38%, and 2.48%, respectively. The greatest advancement is achieved by the IDEA-enhanced PGExplainer, whose average performance (0.8856) even slightly exceeds the counterpart of the current leading baseline ProxyExplainer (0.8690). For ProxyExplainer that already exhibits strong performance, IDEA can further advance its explanation capacity. The IDEA-enhanced ProxyExplainer provides the best explanations for the Alkane-Carbonyl and Fluoride-Carbonyl datasets, among all the evaluated explainers. The sole exception occurs with the IDEA-enhanced V-InFoR on the Mutagenicity dataset, where the explanation performance drops by 5.61%. The possible reason for the degradation and the marginal improvement of IDEA-enhanced V-InFoR is that V-InFoR is specialized for structurally corrupted graphs, while the evaluated graphs are uncorrupted.

## 4.4 ABLATION STUDY (RQ3)

In this section, we probe into the influence of each component in the IDEA framework. First, we replace the Wasserstein distance in Eq.11 with the Kullback-Leibler divergence $\mathrm{KL}[P_g\|P'_G]$ and denote the variant as *IDEA-KL*. Afterwards, to validate the effectiveness of the Explanatory Prototype Alignment stage, we implement two variants, *ID-MSE* and *ID-InfoNCE*, which optimize the GNN explainer by aligning the purified representation of input graph $H'_G$ and the explanation representation $H_g$. *ID-MSE* adopts the mean square error $\mathrm{MSE}(H_g, H'_G)$ as the loss function, and *ID-InfoNCE* adopts the InfoNCE loss function (He et al., 2020a) for in-batch contrastive learning. At last, we omit the Structural Information Disentanglement stage and denote the variant as *EA*. The evaluation result of IDEA and four variants is presented in Figure 4.

We can draw the following conclusions according to the ablation result. First, the distributional discrepancy caused by the explanation subgraph deteriorates the explanation performance. By unifying the representations of the input graph and the explanation subgraph, IDEA and *IDEA-KL* significantly surpass the two variants *ID-MSE* and *ID-InfoNCE* that straightforwardly align the disunified

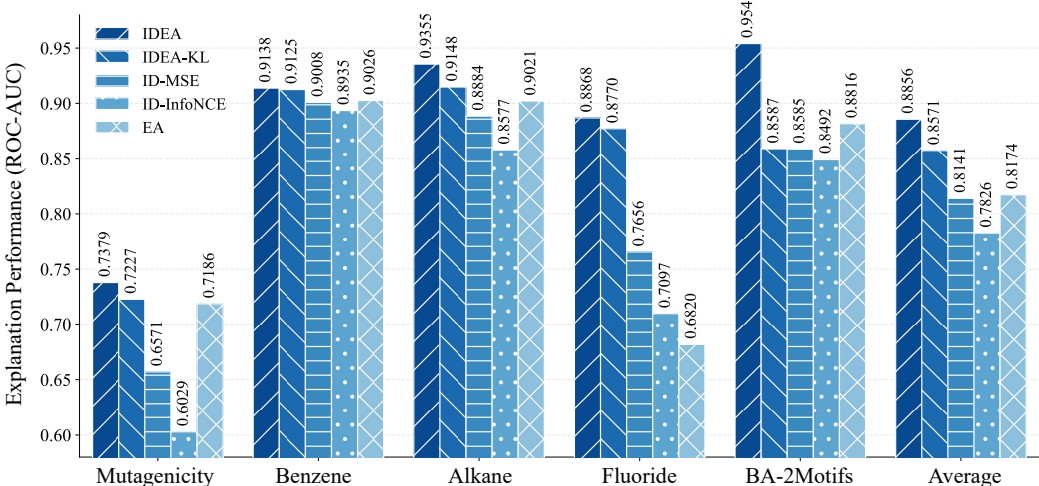

Figure 4: Explanation performance (ROC-AUC ↑) of IDEA and its four variants.

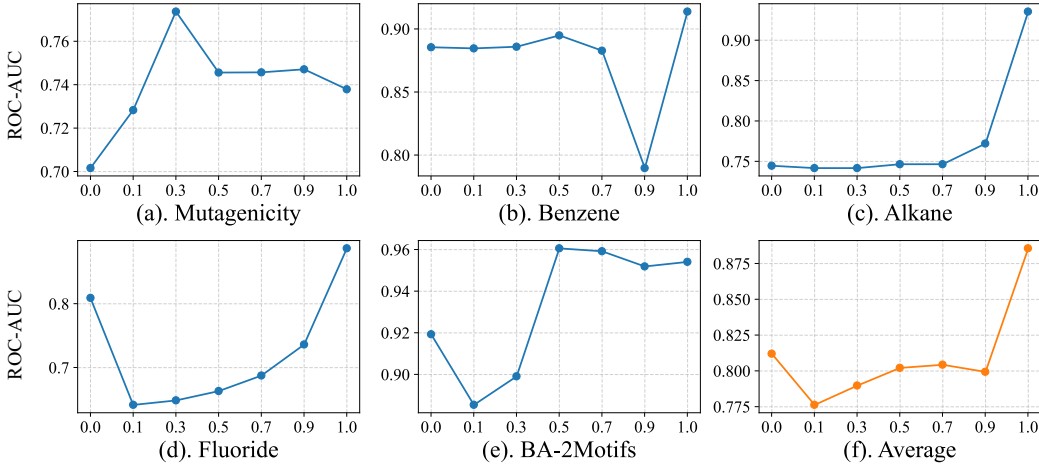

Figure 5: Explanation performance (ROC-AUC ↑) of the combination of the IDEA objective and the label preserving objective, on Mutagenicity, Benzene, Alkane, Fluoride, and BA-2Motifs.

representations. Second, although *EA* is a competitive baseline, structural information disentanglement can further boost the explanation performance. *EA* is inferior to the unabridged IDEA, with an average performance gap by 0.0682. Third, compared with KL divergence, Wasserstein distance is more suitable for GNN explainer optimization in the prototypical representation space. IDEA consistently outperforms *IDEA-KL*, with an average improvement of 3.33%.

**Weighted Combination**. As a natural expansion, we integrate IDEA with the label preserving framework, and the mixed optimization objective is defined as the convex combination as follows,

$$\mathcal{L}_{\text{Mix}} = \alpha \cdot \mathcal{L}_{\text{IDEA}} + (1 - \alpha) \cdot \mathcal{L}_\psi,\ 0 \leq \alpha \leq 1, \tag{14}$$

where $\mathcal{L}_\psi$ denotes the label preserving loss. Typically, $\mathcal{L}_\psi$ is defined as the mutual information between the predictions of the input graph and the explanation subgraph, i.e., $\text{MI}(\hat{y}, \hat{y}_g)$. The evaluation results in Figure 5 reveal that the effectiveness of the integration depends on the specific dataset. For the Mutagenicity dataset, the integration with $\alpha$ equals 0.3 achieves a significant improvement over both IDEA and the label preserving framework. However, for the Benzene, Alkane, and Fluoride datasets, the weighted integration is inferior to both IDEA and the label preserving framework, which might be caused by the counteractive effect between the two optimization objectives.

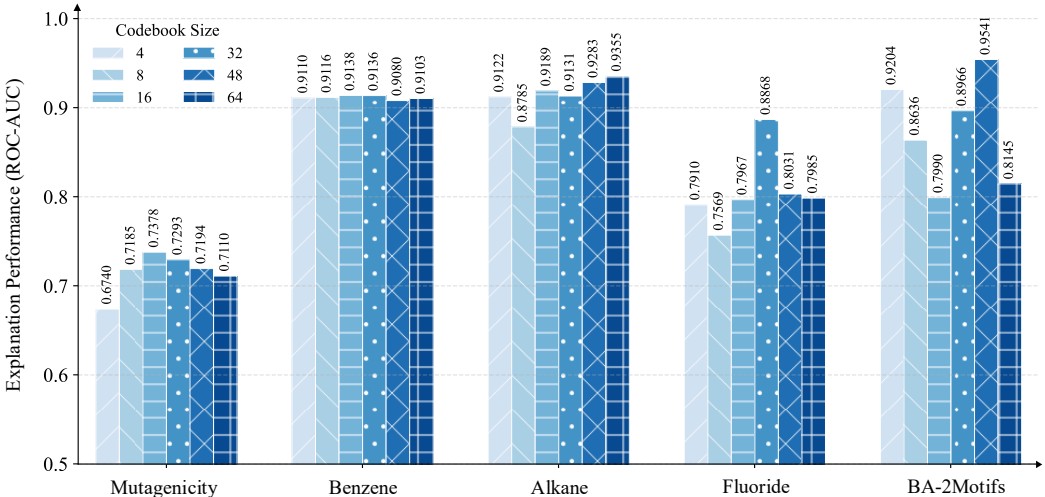

Figure 6: Explanation performance (ROC-AUC ↑) versus the codebook size in IDEA framework, on Mutagenicity, Benzene, Alkane, Fluoride, and BA-2Motifs datasets.

### 4.5 INVESTIGATION ON CODEBOOK SIZE

We further investigate the impact of the codebook size, i.e., the number of codewords within the graph quantizer. As shown in Figure 6, the codebook size ranges among $\{4, 8, 16, 32, 48, 64\}$. In general, a codebook of appropriate size can improve the explanation performance, since it serves as the foundation during structural information disentanglement and explanatory prototype modeling. For the Fluoride dataset, a codebook consisting of 8 codewords causes a performance degradation of 0.1299. Empirically, the recommended value set of $K$ is $\{16, 32, 48\}$, where IDEA achieves the best performance on most of the evaluated datasets.

## 5 RELATED WORK

**Post-hoc Instance-level GNN Explainers** have become a primary approach to explain GNNs, with various methods proposed to identify the critical substructures responsible for predictions. GNNExplainer (Ying et al., 2019) perturbs graph components to estimate their importance. PGExplainer (Luo et al., 2020) introduces a parametric generator to capture global explanatory signals. GraphMask (Schlichtkrull et al., 2021) and ReFine (Wang et al., 2021) advance explanations through edge selection and multi-grained analysis, respectively. D4Explainer (Chen et al., 2023) adopts diffusion models to generate explanations from random noise. V-InFoR (Wang et al., 2023b) and ProxyExplainer (Chen et al., 2024) employ variational autoencoders to enhance explanation robustness.

**Prototype-based GNN explanation methods** aim to improve the intrinsic interpretability of GNNs. ProtGNN (Zhang et al., 2022) introduces prototype learning into GNNs, enabling prototypical subgraphs to serve as intuitive analogical explanations. PAGE (Shin et al., 2024) extends this idea to model-level interpretability by constructing a latent prototype dictionary, offering explanations of the overall decision boundary. Dai & Wang (2025) further integrate prototype learning with self-explaining mechanisms, jointly optimizing prediction and interpretability.

## 6 CONCLUSION

We, for the first time, propose the paradigm shift of the GNN explainer optimization framework from the graph label space to the graph representation space, and we design IDEA, the first GNN explainer optimization framework grounded in a prototypical graph representation space. IDEA consists of a structural information disentanglement stage, which disentangles and encapsulates the explanatory substructures into prototypes, and an explanatory prototype alignment stage, which aligns the representational distributions of the input graph and the explanation unified in the prototypical space. Extensive experiments demonstrate the effectiveness and generalizability of IDEA.

ACKNOWLEDGEMENT

J Yin and S Wang are supported by the National Natural Science Foundation of China (No.62572489). C Li is supported by the Beijing Natural Science Foundation (No.L251037) and the Fundamental Research Funds for the Central Universities. In particular, this work is carried out using computing resources supported by HyperAI.

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

# A  NOTATION

In Table 4, we summarize the notations used throughout this manuscript and their descriptions.

Table 4: Notations and corresponding descriptions.

| Notation | Description |
|---|---|
| $G$ | Graph instance |
| $A, X$ | Adjacency matrix, node feature matrix |
| $N$ | Number of graph nodes |
| $D$ | Node feature dimension |
| $v_i$ | The $i$-th node |
| $A_{ij}$ | Element at the $i$-th row, $j$-th column of adjacency matrix $A$ |
| $y, \hat{y}$ | Graph label, GNN prediction |
| $C$ | Number of graph classes |
| $f, f_e, f_p$ | Graph neural network model, encoder of $f$, predictor of $f$ |
| $H_N, H_G$ | Node representation matrix, graph representation vector |
| $h_i$ | Node representation of $v_i$ |
| $\psi$ | Post-hoc instance-level GNN explainer |
| $g, g^*$ | Explanatory subgraph (i.e., explanation) |
| $\mathcal{L}_\psi$ | Label preserving loss of $\psi$ |
| $\Omega$ | Regularization term |
| $\lambda_\Omega$ | Weighted hyper-parameter of $\Omega$ |
| $\mathrm{GQ_S}, \mathrm{GQ_D}$ | Shallow graph quantizer, deep graph quantizer |
| $\mathcal{C}_S, \mathcal{C}_D$ | Codebook of $\mathrm{GQ_S}$, codebook of $\mathrm{GQ_D}$ |
| $\mathcal{D}$ | Distance metric of vector quantization |
| $q, q^*$ | Codeword, the nearest codeword |
| $q_S^*, q_D^*$ | The nearest codeword in $\mathrm{GQ_S}$, the nearest codeword in $\mathrm{GQ_D}$ |
| $h_i'$ | Residual representation after $\mathrm{GQ_S}$ quantization |
| $\mathcal{L}_Q, \mathcal{L}_S, \mathcal{L}_D$ | Quantization objective, structure-aware objective, disentanglement objective |
| $Q^*$ | Quantization matrix |
| $\sigma$ | Sigmoid function |
| $f_d$ | Linear decoder |
| $\hat{y}_S$ | GNN prediction of $\mathrm{GQ_S}$ quantized representation |
| $\hat{y}_D$ | GNN prediction of $\mathrm{GQ_D}$ quantized representation |
| $\mathcal{U}_C$ | Uniform distribution |
| $\mathcal{L}_{\mathrm{SAD}}$ | Structure-aware disentanglement objective |
| $\lambda_Q, \lambda_S$ | Weighted hyper-parameter of $\mathcal{L}_Q$ and $\mathcal{L}_S$ |
| $H_G, H_g$ | Representation of original graph, representation of explanation |
| $H_G'$ | Residual representation of $H_o$ after $\mathrm{GQ_S}$ quantization |
| $\mathcal{P}_G', \mathcal{P}_g$ | Assignment probability of $H_o'$ and $H_e$ representation of explanation |
| $W_\epsilon$ | Entropy-regularized Wasserstein distance |
| $\Pi, S$ | Transport polytope and cost matrix of $W_\epsilon$ |
| $\mathcal{L}_{\mathrm{IDEA}}$ | IDEA optimization objective |
| $\mathcal{L}_{\mathrm{Mix}}$ | Weighted combination of $\mathcal{L}_\psi$ and $\mathcal{L}_{\mathrm{IDEA}}$ |
| $\alpha$ | Weighted parameter in $\mathcal{L}_{\mathrm{Mix}}$ |

# B  EXPERIMENTAL DETAIL

Here, we elaborate the details of the evaluated datasets, baselines, and IDEA implementation.

## B.1  DATASET

The dataset details are introduced as follows and the dataset statistics are summarized in Table 5.

- **Mutagenicity** (Kazius et al., 2005) is a collection of molecular compounds labeled for their ability to cause mutations (i.e., mutagenic vs. non-mutagenic), widely used in cheminformatics and toxicology for developing predictive models. Mutagenicity contains 4,337 molecule graphs with $NO_2$ and $NH_2$ chemical groups notated as ground truth explanations.

Table 5: The statistics of the evaluated datasets.

| Statistic | Mutagenicity | Benzene | Alkane | Fluoride | BA-2Motifs |
|---|---|---|---|---|---|
| Graphs | 4,337 | 12,000 | 4,326 | 8,671 | 1,000 |
| Average Nodes | 30.32 | 20.58 | 21.13 | 21.36 | 25.00 |
| Average Edges | 30.77 | 43.65 | 44.95 | 45.37 | 50.90 |
| Node Features | 14 | 14 | 14 | 14 | 10 |
| GNN Accuracy | 0.8300 | 0.9054 | 0.9620 | 0.9340 | 1.0 |
| GT Explanation | $NO_2$, $NH_2$ | Benzene | Alkane + C=O | $F^-$ + C=O | Motif |

- **Benzene** (Agarwal et al., 2023) is a binary classification dataset of 12,000 molecular graphs sampled from ZINC15 (Sterling & Irwin, 2015). The goal is to decide whether a molecule contains at least one benzene ring. Within this dataset, the atoms that constitute the benzene ring serve as the ground-truth explanation. Multiple disjoint benzene rings are treated as separate explanations.

- **Alkane-Carbonyl** (Agarwal et al., 2023) is a binary classification set of 4,326 molecular graphs. A positive label marks a molecule that simultaneously contains an unbranched alkane chain and a carbonyl (C=O) group. The ground-truth explanation is defined as the arbitrary union of these two functional fragments present in the structure.

- **Fluoride-Carbonyl** (Agarwal et al., 2023) contains 8,671 molecular graphs. A molecule is labeled positive only if it contains both a fluoride atom (F) and a carbonyl group (C=O). The explanation is defined as the arbitrary union of these two functional units found in the structure.

- **BA-2Motifs** (Ying et al., 2019) is a synthetic binary class dataset designed to benchmark GNN explanation methods. Each graph is labeled by the presence of either a house or a cycle motif, and the respective motif itself provides the ground truth explanation for that class.

We present the accuracy of the to-be-explained GNN model for each dataset in Table 5 as well.

## B.2 BASELINE

The evaluated baselines include eight SOTA post-hoc instance-level GNN explainers based on various search strategies. The details are introduced as follows.

- **GNNExplainer** (Ying et al., 2019) is a GNN explainer based on data perturbation that jointly masks edges and node features, then scores their contribution by searching for a subgraph $G_S$ that maximizes the mutual information with the model's overall prediction $\hat{y}$.

- **PGExplainer** (Luo et al., 2020) masks graph topology and uses a learnable neural network to assign edge importance scores, optimizing the same mutual-information objective.

- **GraphMask** (Schlichtkrull et al., 2021) learns an amortized classifier that predicts whether the edge can be dropped (replaced by a learned baseline) for every edge in every GNN layer, without changing the model output, yielding a sparse post-hoc explanation.

- **ReFine** (Wang et al., 2021) adopts a pre-train and fine-tune strategy to probe GNN decisions, delivering multi-granularity insights into the model's reasoning process.

- **V-InfoR** (Wang et al., 2023b) is a robust GNN explainer specialized for the structurally corrupted graphs, which employs the variational inference to learn the robust graph representations and generalizes the GNN explanation exploration to a graph information bottleneck (GIB) optimization task without any predefined rigorous constraints.

- **D4Explainer** (Chen et al., 2023) a generative explainer for counterfactual and model-level explanations based on a discrete denoising diffusion model, which frames the explanation problem as a distribution learning task for more reliable explanations with better in-distribution property.

- **MixupExplainer** (Zhang et al., 2023) addresses the distribution shifting issue by mixing up the explanation with a randomly sampled base graph structure.

- **ProxyExplainer** (Chen et al., 2024) extends the GIB by innovatively including in-distributed proxy graphs and derives a tractable objective function for practical implementations, where two graph auto-encoders are utilized to generate proxy graphs.

### B.3 IDEA IMPLEMENTATION

Within the main experiment, we adopt a well-investigated subgraph generator as the backbone implementation of the IDEA framework. According to the Gilbert random graph theory, an arbitrary graph $G$ can be represented as a random graph variable, and each edge of $G$ is associated with a binary random variable $r$ to reveal its existence. Additionally, the existence of one edge is conditionally independent of the other edges. $\varepsilon_{ij} = 1$ means there is an edge $(i, j)$ from $v_i$ to $v_j$, otherwise $\varepsilon_{ij} = 0$. Hence, an arbitrary graph $G$ can be represented as follows,

$$p(G) = \prod_{(i,j)} p(\varepsilon_{ij}). \tag{15}$$

A common instantiation of the binary variable $\varepsilon_{ij}$ is the Bernoulli distribution $\varepsilon_{ij} \sim \text{Bern}(\varrho_{ij})$, where $\varrho_{ij} = p(\varepsilon_{ij} = 1)$ is the probability of edge $(i, j)$ existing in the random graph $G$. Since the Bernoulli distribution cannot be directly optimized, we introduce categorical reparameterization (Jang et al., 2017) to $\varepsilon_{ij}$. The continuous relaxation of $\varepsilon_{ij}$ can be formulated as follows,

$$\varepsilon_{ij} = \sigma\Big(\frac{\log \mathcal{U} - \log\left(1 - \mathcal{U}\right) + \mu_{ij}}{\tau}\Big), \; \mu_{ij} = \log\frac{\varrho_{ij}}{1 - \varrho_{ij}}, \; \mathcal{U} \sim \text{Uniform}(0, 1). \tag{16}$$

where $\tau$ controls the approximation between the relaxed distribution and $\text{Bern}(\varrho_{ij})$. When $\tau$ approaches 0, the limitation of Eq.(16) is $\text{Bern}(\varrho_{ij})$.

According to Eq.(16), the Bernoulli parameter $\varrho_{ij}$ is associated with the parameter $\mu_{ij}$. To enable end-to-end optimization, we use a multi-layer perceptron (MLP) to compute $\mu_{ij}$. The MLP takes the GNN node representation as input and concatenates the representations of two nodes $v_i, v_j$ as the representation of the corresponding edge $(i, j)$, which can be formulated as follows,

$$\mu_{ij} = \text{MLP}\big([h_i \| h_j]\big), \tag{17}$$

where $[\cdot \| \cdot]$ is the concatenation operator. Based on $\mu = \{\mu_{ij} | i, j = 1, 2, \cdots, N\}$ and Eq.(16), we obtain the probability matrix $M_\mu$ whose elements indicate the existence of the corresponding edges. Afterwards, we can sample the explanation $g$ based on the probabilities in the matrix $M_\mu$ as follows,

$$g = \big(X_S, A_S = M_\mu \odot A\big). \tag{18}$$

So far, we have derived the optimizable representation of $g$ utilized in IDEA. All experiments are finished on a machine with 4 *NVIDIA GeForce RTX 3090 24GiB* GPUs.

### B.4 STEP-BY-STEP BREAKDOWN OF HGTOKENIZER

Given the input embedding $h_i \in \mathbb{R}^{1 \times d}$, the shallow codebook $\mathcal{C}_S \in \mathbb{R}^{K \times d}$, and the deep codebook $\mathcal{C}_D \in \mathbb{R}^{K \times d}$, the details of HGTokenizer process are elaborated as follows.

- Step 1. Feed the input embedding $h_i$ into the shallow quantizer. The shallow quantizer first calculates the pair-wise distance between $h_i$ and each codeword $q \in \mathbb{R}^{1 \times d}$ within the shallow codebook $\mathcal{C}_S$. Afterwards, the shallow quantizer select the closest codeword to $h_i$ according to the $K$-dimensional distance vector. Step 1 is formulated by the formula $q_{S,i}^* = \text{GQ}_S(h_i) = \arg\min_{q \in \mathcal{C}_S} \mathcal{D}(q, h_i)$ in Equation 2.
- Step 2. Calculate the quantization residual of the shallow quantizer, which is formulated by the formula $h_i' = h_i - q_{S,i}^*$ in Equation 2.
- Step 3. Feed the residual embedding $h_i'$ into the deep quantizer, the detailed quantization process is the same as that in Step 1. The deep quantizer will select the closest codeword $q_{D,i}^* = \text{GQ}_D(h_i') = \arg\min_{q \in \mathcal{C}_D} \mathcal{D}(q, h_i')$, as shown in Equation 3.

Subsequently, the quantized representations $q_{S,i}^*$ and $q_{D,i}^*$ provide by the shallow and deep quantizers are used to compute the disentanglement loss $\mathcal{L}_D$. The sum of them, i.e., $q_i^* = q_{S,i}^* + q_{D,i}^*$ in Equation 3, is used to compute the structure-awareness loss $\mathcal{L}_S$.

## C HYPER-PARAMETER ANALYSIS

In Table 6, we summarize the optimal configuration of hyper-parameters in IDEA for each dataset.

Table 6: The optimal configuration of hyper-parameters in IDEA.

| Hyper-parameter | Mutagenicity | Benzene | Alkane | Fluoride | BA-2Motifs |
|---|---|---|---|---|---|
| ID Epochs | 10 | 10 | 5 | 5 | 15 |
| ID Learning Rate | 0.01 | 0.01 | 0.005 | 0.0005 | 0.001 |
| EA Epochs | 20 | 10 | 15 | 10 | 10 |
| EA Learning Rate | 0.0001 | 0.0005 | 0.001 | 0.001 | 0.0001 |
| Batch Size | 20 | 64 | 64 | 32 | 20 |
| Codebook Size | 16 | 32 | 64 | 32 | 48 |

Figure 7: Explanation performance (ROC-AUC ↑) versus the weighted parameter $\lambda_Q$ ($y$-axis) and $\lambda_S$ ($x$-axis) in the SAD objective, on (a). Mutagenicity, (b). Benzene, (c).Alkane, and (d). Fluoride.

## C.1 STRUCTURE-AWARE DISENTANGLEMENT OBJECTIVE

Here, we investigate the impact of the weighted parameters $\lambda_Q$ and $\lambda_S$ in the strcuture-aware disentanglement objective defined as Eq.7,

$$\mathcal{L}_{\text{SAD}} = \mathcal{L}_D + \lambda_S \cdot \mathcal{L}_S + \lambda_Q \cdot \mathcal{L}_Q.$$

The evaluated result is presented in Figure 7. As the weighted hyper-parameters range from 0.1 to 5.0, we can notice that the optimal performance is more likely to be achieved when the objective weights are balanced, i.e., along the diagonal direction of the heatmap. For the Mutagenicity and Benzene datasets, the best performance is achieved by setting $\lambda_S = \lambda_Q = 1.0$. For the Alkane and Fluoride datasets, the best configurations of weighted parameters are $\lambda_S = 0.5, \lambda_Q = 0.1$ and $\lambda_S = 0.1, \lambda_Q = 0.5$, without severe unbalance.

## D EXPLANATION VISUALIZATION

From Figure 8 to Figure 11, we present the explanation visualization of the ground truth, IDEA, and four SOTA GNN explainers based on the label preserving framework.

Table 7: Runtime (Second ↓) of four native explainers with different architectures and the corresponding IDEA variants. *Times* is defined as `IDEA/Native`.

| Model | Mutagenicity | Benzene | Alkane | Fluoride | Average |
|---|---|---|---|---|---|
| PGExplainer | 4.92 | 3.80 | 2.24 | 6.67 | 4.41 |
| +IDEA | 14.53 | 10.34 | 5.43 | 19.13 | 12.36 |
| *Times* | 2.95 | 2.72 | 2.42 | 2.87 | 2.80 |
| ReFine | 13.14 | 39.44 | 13.21 | 28.10 | 23.47 |
| +IDEA | 21.04 | 61.90 | 20.29 | 50.45 | 38.42 |
| *Times* | 1.60 | 1.57 | 1.54 | 1.80 | 1.64 |
| V-InFoR | 5.31 | 11.70 | 7.17 | 18.69 | 10.72 |
| +IDEA | 13.66 | 29.19 | 19.25 | 45.48 | 26.90 |
| *Times* | 2.57 | 2.49 | 2.68 | 2.43 | 2.51 |
| ProxyExplainer | 20.90 | 15.23 | 7.33 | 14.82 | 14.57 |
| +IDEA | 21.52 | 16.52 | 7.90 | 15.85 | 15.45 |
| *Times* | 1.03 | 1.08 | 1.08 | 1.07 | 1.06 |

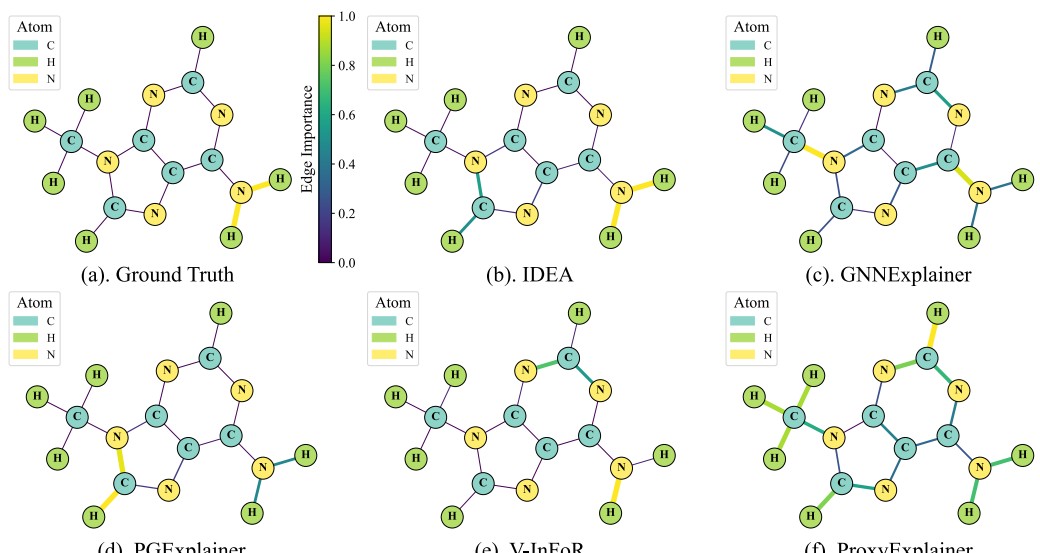

(a). Ground Truth     (b). IDEA     (c). GNNExplainer

(d). PGExplainer     (e). V-InFoR     (f). ProxyExplainer

Figure 8: Explanation visualization of ground truth, IDAE, and four baselines on Mutagenicity.

As shown in Figure 8, only IDAE assigns the highest importance score to $NH_2$. PGExplainer and ProxyExplainer assign medium scores to $NH_2$, V-InFoR identifies part of the $NH_2$ group, and GNNExplainer fails to detect $NH_2$. In Figure 9, IDEA, PGExplainer, and ProxyExplainer successfully identify the benzene ring. In particular, IDEA detects the two rings within the molecule, while PGExplainer and ProxyExplainer notice only part of the second benzene ring. GNNExplainer and V-InFoR fail to assign high scores to the benzene rings. For the Alkane dataset, IDEA and GNNExplainer identify the chlorine atom Cl as the explanation, yet the other three explainers completely ignore the influential substrcutures. In Figure 11, only IDEA and V-InFoR can discriminate the explanatory structure from the confounding structures to some extent. The other three explainers assign nearly identical scores to all edges.

# E   TIME COMPLEXITY

In this section, we first provide a theoretical analysis of the time complexity of the IDEA framework. Then, we report the runtime of diverse explainer architectures, incorporating with both the IDEA framework and the label preserving framework. Given the node presentation matrix $H_N \in \mathbb{R}^{N \times d}$, HGTokenizer approximates it by two graph quantizers, including two matrix multiplication opera-

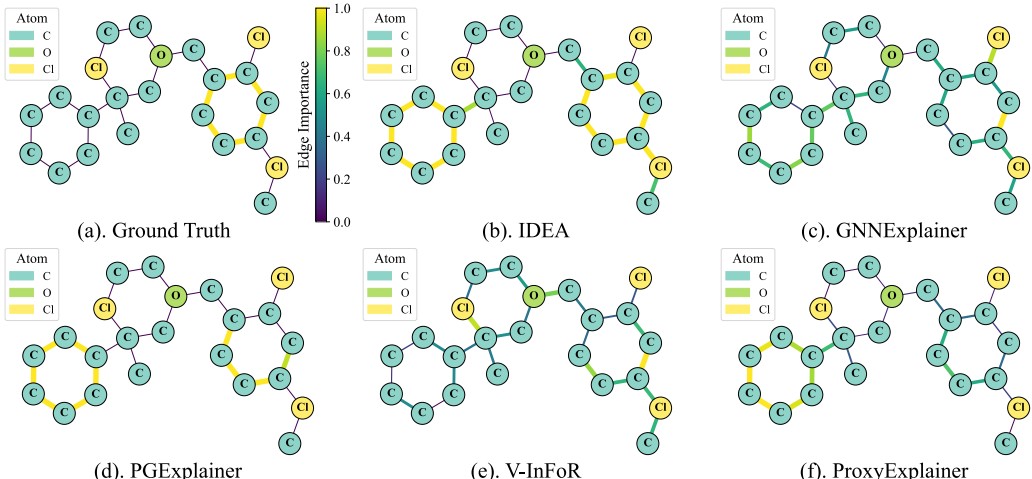

Figure 9: Explanation visualization of ground truth, IDAE, and four baselines on Benzene.

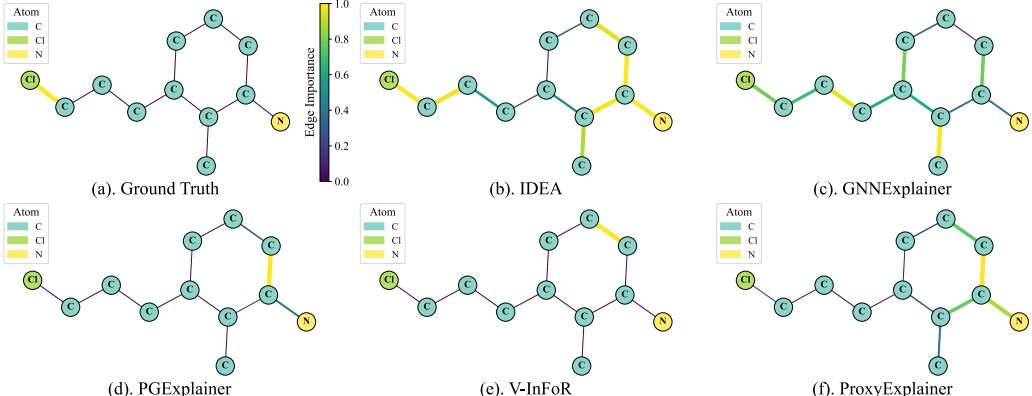

Figure 10: Explanation visualization of ground truth, IDAE, and four baselines on Alkane.

tions and two `argmin` operations. The codebook within the graph quantizer belongs to a matrix in $\mathbb{R}^{K \times d}$. Hence, the time complexity of quantization distance $\mathcal{D}$ is

$$\vartheta_{\mathcal{D}} = NKd. \tag{19}$$

Since the time complexity of `argmin` is $NK$, the total complexity of HGTokenizer is

$$\vartheta_{\text{HGT}} = 2NK(d+1). \tag{20}$$

Therefore, the complexity of IDEA, including the structural information disentanglement and the explanatory prototype alignment, is derived as,

$$\vartheta_{\text{IDEA}} = 2NK(d+1) + 3K(d+1) = O(NKd). \tag{21}$$

According to Eq.21, the time complexity is linear to the node number of input graph, the codebook size, and the hidden dimension of target GNN.

In Table 7, we report the runtime of four different GNN explainers and their counterparts shifted to the IDEA framework. One can note that the runtime of IDEA variants is of the same magnitude, compared with the native explainer adopting the label preserving framework.

## F SUPPLEMENTARY EXPERIMENT

### F.1 CONJOINT OPTIMIZATION OF IDEA

In our main experiment, IDEA is a dual-stage framework, where the Structural Information Disentanglement and the Explanatory Prototype Alignment are conducted separately. The dual stage

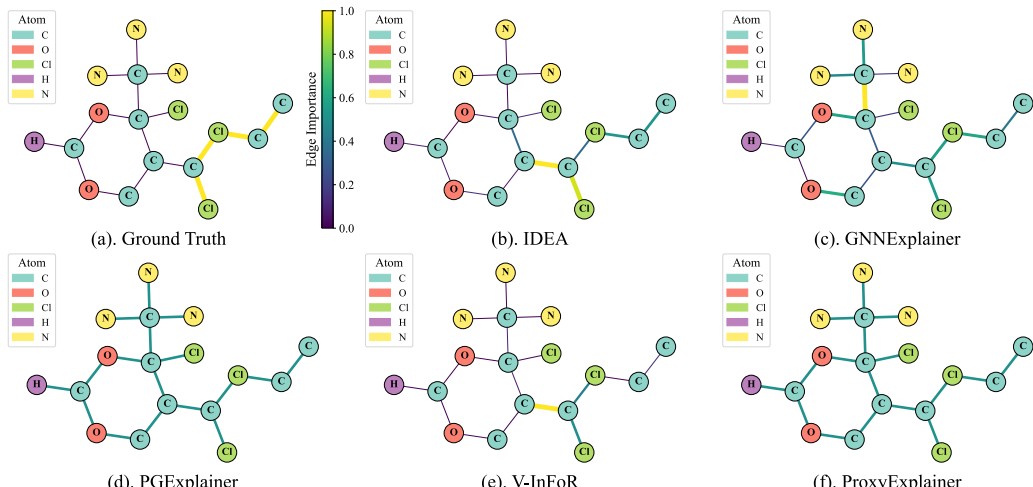

Figure 11: Explanation visualization of ground truth, IDAE, and four baselines on Fluoride.

Table 8: Explanation performance (ROC-AUC ↑) of IDEA and the conjoint variant.

| Model | Mutagenicity | Benzene | Alkane | Fluoride | Average |
|---|---|---|---|---|---|
| IDEA | 0.7379 | 0.9138 | 0.9319 | 0.8868 | 0.8676 |
| IDEA-Joint | 0.4805 | 0.5447 | 0.8725 | 0.8349 | 0.6832 |

implementation not only reduces the difficulty of IDEA optimization, but also avoids the counteraction effect between the optimization objectives. To empirically validate the rationality of dual-stage IDEA, we further implement an IDAE variant *IDEA-Joint* where the two stages are conducted jointly. The optimization objective of *IDEA-Joint* is defined as follows,

$$\mathcal{L}_{\text{Joint}} = \mathcal{L}_{\text{IDAE}} + \lambda_{\text{SAD}} \cdot \mathcal{L}_{\text{SAD}}, \tag{22}$$

with $\mathcal{L}_{\text{IDEA}}$ and $\mathcal{L}_{\text{SAD}}$ defined by Eq.11 and Eq.7, respectively. In Table 8, we present the performance comparison between IDEA and the conjoint variant, within the same hyper-parameter search range. We can notice the evident gap between *IDEA-Joint* and IDEA, which implies the difficulty of *IDEA-Joint* optimization, despite a possible performance upper bound better than IDEA.

## F.2    Robustness to Label Noise

To investigate the robustness of the IDEA explainer to label noise (Zhong et al., 2023), we perturb the information disentanglement stage by flipping the GNN prediction $\hat{y}$ in Eq.5 and present the result in Table 9. For comparison, we evaluate the explanation performance of two typical explainers, i.e., GNNExplainer (Ying et al., 2019) and PGExplainer (Luo et al., 2020), with the same setting of label noise. Specifically, the intensity of the label noise ranges from 0.00% to 50.00%, with an interval of 5.00%. As shown by the result, the IDEA explainer stably maintains the high quality of the generated explanation, with the maximum performance degradation of 0.0076 and 0.0070 in the Mutagenicity and Benzene datasets, respectively. In contrast, two typical GNN explainers based on the label preserving framework, GNNExplainer and PGExplainer, suffer from severe performance degradation, which is 14.03× times and 22.58× times greater than that of IDEA.

## G    Theoretical Justification

During the explanatory prototype alignment stage, we adopt the assignment probability of the input representation ($H'_G$ or $H_g$) over the explanatory codebook $\mathcal{C}_D$ to reflect its location within the prototypical representation space. In this section, we elaborate the justification of this practice. Within the explanatory codebook $\mathcal{C}_D$, we have $K$ prototype codewords $\{q_1, q_2, \cdots, q_K\} \subset \mathbb{R}^d$, which expand the prototypical representation space. Taking the prototype codewords $\{q_1, q_2, \cdots, q_K\}$ as the

Table 9: Explanation performance (ROC-AUC ↑) versus label noise intensity on Mutag and Benzene datasets. $\Delta_{\max}$ presents the maximum performance degradation with noise intensity increasing.

| Noise Intensity | Mutagenicity | | | Benzene | | |
|---|---|---|---|---|---|---|
| | GNNExplainer | PGExplainer | IDEA | GNNExplainer | PGExplainer | IDEA |
| 0.00% | 0.6155 | 0.7016 | **0.7379** | 0.6886 | 0.8855 | **0.9138** |
| 5.00% | 0.6140 | 0.6989 | **0.7358** | 0.6662 | 0.8856 | **0.9128** |
| 10.00% | 0.6063 | 0.6824 | **0.7359** | 0.6505 | 0.8856 | **0.9135** |
| 15.00% | 0.5937 | 0.6819 | **0.7363** | 0.6326 | 0.8860 | **0.9139** |
| 20.00% | 0.5954 | 0.6810 | **0.7320** | 0.6149 | 0.5784 | **0.9132** |
| 25.00% | 0.5965 | 0.6805 | **0.7366** | 0.5966 | 0.5931 | **0.9128** |
| 30.00% | 0.6050 | 0.6802 | **0.7319** | 0.5788 | 0.5932 | **0.9128** |
| 35.00% | 0.6048 | 0.6801 | **0.7303** | 0.5636 | 0.6302 | **0.9132** |
| 40.00% | 0.6065 | 0.6798 | **0.7366** | 0.5431 | 0.6591 | **0.9131** |
| 45.00% | 0.6048 | 0.6795 | **0.7327** | 0.5282 | 0.7033 | **0.9068** |
| 50.00% | 0.6058 | 0.6791 | **0.7328** | 0.5056 | 0.7340 | **0.9130** |
| $\Delta_{\max}$ ↓ | 0.0218 | 0.0225 | **0.0076** | 0.1830 | 0.3071 | **0.0070** |

anchors, the $L_2$ distance between the input representation $h$ and the anchor $q_k$ is

$$\varphi_k = \|h - q_k\|^2 = h^T h + q_k^T q_k - 2q_k^T h. \tag{23}$$

For $k \geq 2$, by subtracting $\varphi_1 = \|h - q_1\|^2$, we can derive the following equation

$$\varphi_k - \varphi_1 = (q_k^T q_k - q_1^T q_1) - 2(q_k - q_1)^T h, \tag{24}$$

which is equivalent to the equation below,

$$(q_k - q_1)^T h = \frac{1}{2}\Big(q_k^T q_k - q_1^T q_1 + \varphi_1 - \varphi_k\Big). \tag{25}$$

For $k = 2, 3 \cdots, K$, stacking $(q_k - q_1)^T h$ induces the following equation in matrix formulation,

$$\begin{bmatrix} (q_2 - q_1)^T \\ (q_3 - q_1)^T \\ \vdots \\ (q_K - q_1)^T \end{bmatrix} h = \frac{1}{2} \begin{bmatrix} q_2^T q_2 - q_1^T q_1 + \varphi_1 - \varphi_2 \\ q_3^T q_3 - q_1^T q_1 + \varphi_1 - \varphi_3 \\ \vdots \\ q_K^T q_K - q_1^T q_1 + \varphi_1 - \varphi_K \end{bmatrix}, \tag{26}$$

which can be briefly noted as

$$Ah = b, \ A \in \mathbb{R}^{(K-1)\times d}, \ b \in \mathbb{R}^{(K-1)}. \tag{27}$$

Theoretically, the prototype codewords and the induced prototypical representation space are generated by a collection of latent variables $\{z_1, z_2, \cdots, z_t\} \subset \mathbb{R}^{d'}$ with $d' < d$, which objectively determine the explanatory substructures while being unobservable. Therefore, Eq.27 implies a counterpart in the latent space $\mathbb{R}^{d'}$ as follows,

$$A'h' = b', \ A' \in \mathbb{R}^{(K-1)\times d'}, \ b' \in \mathbb{R}^{(K-1)}. \tag{28}$$

In this equation, when $K \geq d' + 1$, $h'$ has a unique solution. Therefore, we adopt the assignment probability based on the quantization distance to indicate the location of the input representation, instead of training an additional projector.

We denote the unknown mapping function from the latent space $\mathbb{R}^{d'}$ to the prototypical space $\mathbb{R}^d$ as $\mathcal{H} : \mathbb{R}^{d'} \to \mathbb{R}^d$. For two representations $x, y \in \mathbb{R}^d$, the corresponding representations in $\mathbb{R}^{d'}$ are denoted as $x' = \mathcal{H}^{-1}(x)$ and $y' = \mathcal{H}^{-1}(y)$. In our method, we minimize the difference between the assignment probabilities of $x$ and $y$ over the prototypes $\{q_1, q_2, \cdots, q_K\}$, in order to minimize the distance between $x'$ and $y'$ in the latent space. Theoretically, the strict validity of this measurement lies in three conditions. First, $K \geq d'$, which holds with large probability. Second, the prototype representations $\{q_1, q_2, \cdots, q_K\}$ are linearly independent. As illustrated in Figure 12, we present the cosine similarity of the codewords, demonstrating that the codewords approximately satisfy the linearly independent requirement. Third, the hypothetical mapping function $\mathcal{H}$ is linear or can be approximated by linear functions.

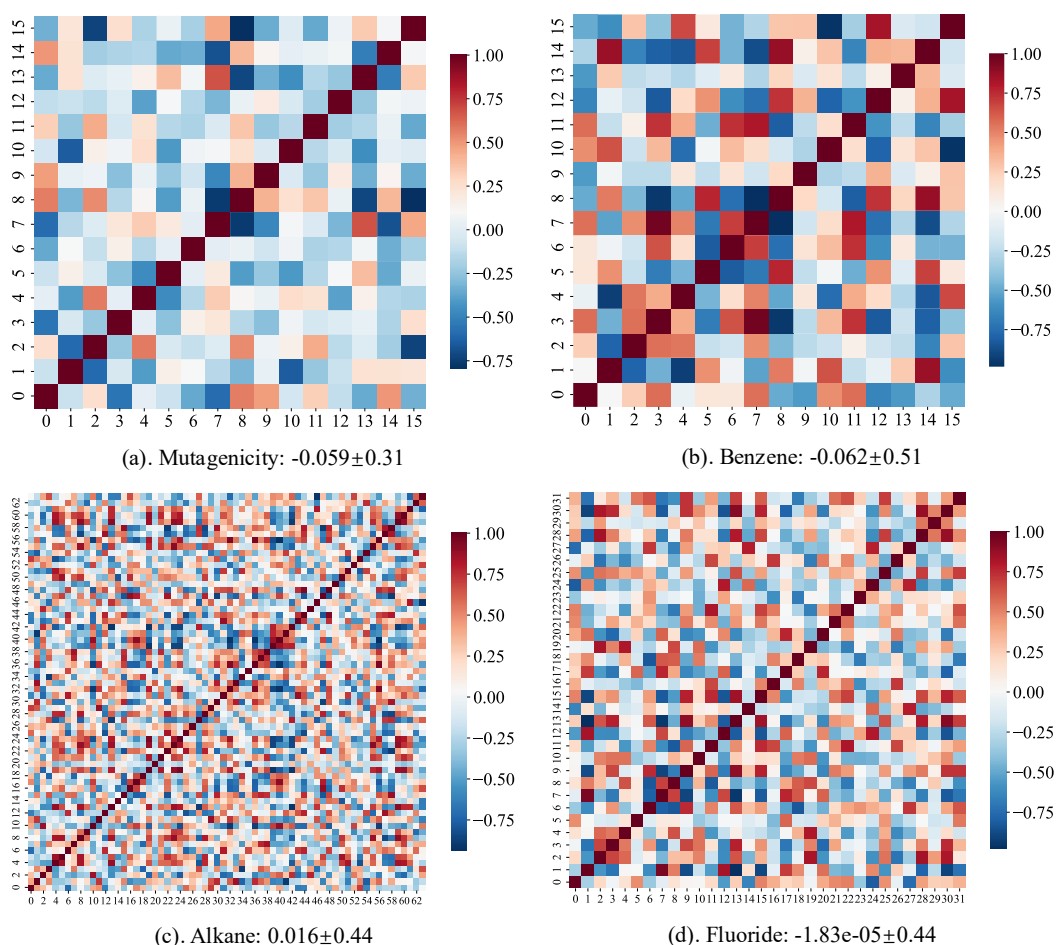

Figure 12: Pair-wised cosine similarity of codewords. The number behind the dataset name represents Mean ± Std.

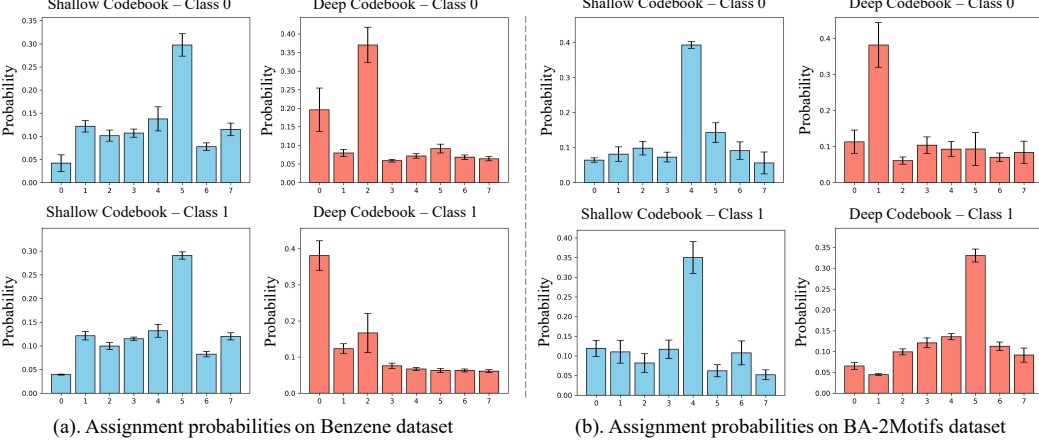

Figure 13: Average probabilistic distributions over the shallow and deep codebooks on (a). Benzene dataset and (b). BA-2Motifs dataset.

## H PROTOTYPE CASE STUDY

**Assignment Probability**. First, to explore the implicit relationship between the prototypical embeddings (i.e., codebooks) and human-intelligible substructures, we present the assignment probabilities

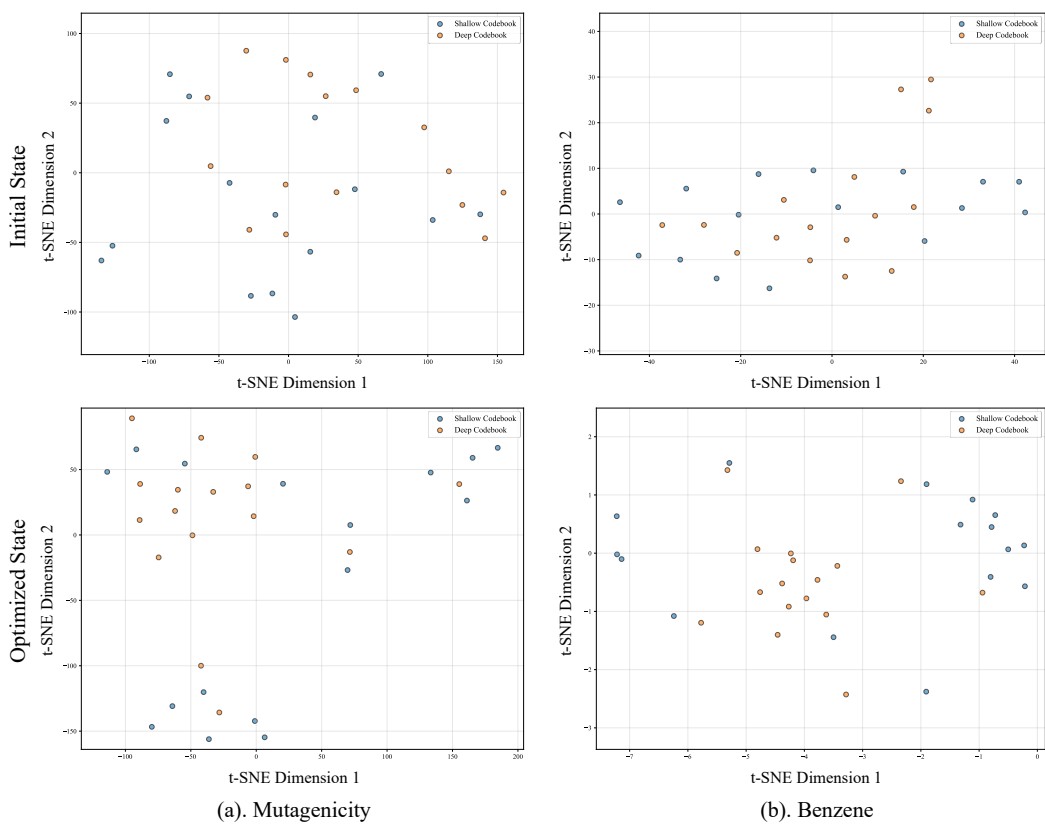

(a). Mutagenicity

(b). Benzene

Figure 14: t-SNE visualization of codewords on (a). Mutagenicity dataset and (b) Benzene dataset.

distribution in Figure 13. Specifically, for real-world dataset Benzene and synthetic dataset BA-2Motifs, we visualize the average probabilistic distributions of class 0 and class 1 over the shallow and deep codebooks. For the real-world dataset Benzene, the distributions of class 0 and class 1 over the shallow codebook are similar, and the codeword 5 with the largest probability may correspond the most frequent non-explanatory substructure (carbon-chlorine bond). On the deep codebook, the distribution patterns obviously differ. For the deep codebook, the codeword 0 may correspond to the benzene rings which directly decides the labels of class 1, and the codeword 2 may correspond to the carbon-oxygen bond which is common in class 0. For the synthetic dataset BA-2Motifs, the shallow distribution patterns of class 0 and class 1 are also similar. The deep shallow distribution has two peaks, i.e., codeword 1 and codeword 5, which may correspond to the two kinds of motifs in BA-2Motifs. To sum up, the similar distribution pattern on shallow codebook and significantly different patterns on deep codebook can indicate that the learned prototypes in codebooks are implicitly related to substructures.

**t-SNE Visualization**. Furthermore, we visualize the learned codewords in shallow and deep codebooks based on t-SNE algorithm (van der Maaten & Hinton, 2008). As shown by Figure 14 and Figure 15, the first row presents the t-SNE visualization of the initial codewords, and the second row presents that of the codewords after optimization, i.e., prototypes. We can notice that in the initial state, the shallow and deep codewords mix together without clear boundary. After optimization, the deep codewords are approximately separable from the shallow ones. The deep codewords prefer to cluster into a mass, while the shallow codewords still distribute dispersedly.

# I LIMITATION

**Accessibility to target GNN**. According to the taxonomy of GNN explanation methods, the accessibility of the GNN explainer to the target GNN to be explained can be categorized into black-box, gray-box, and white-box. The black-box accessibility takes the GNN model as an oracle and only requires the GNN predictions. On the contrary, the white-box accessibility demands the permission

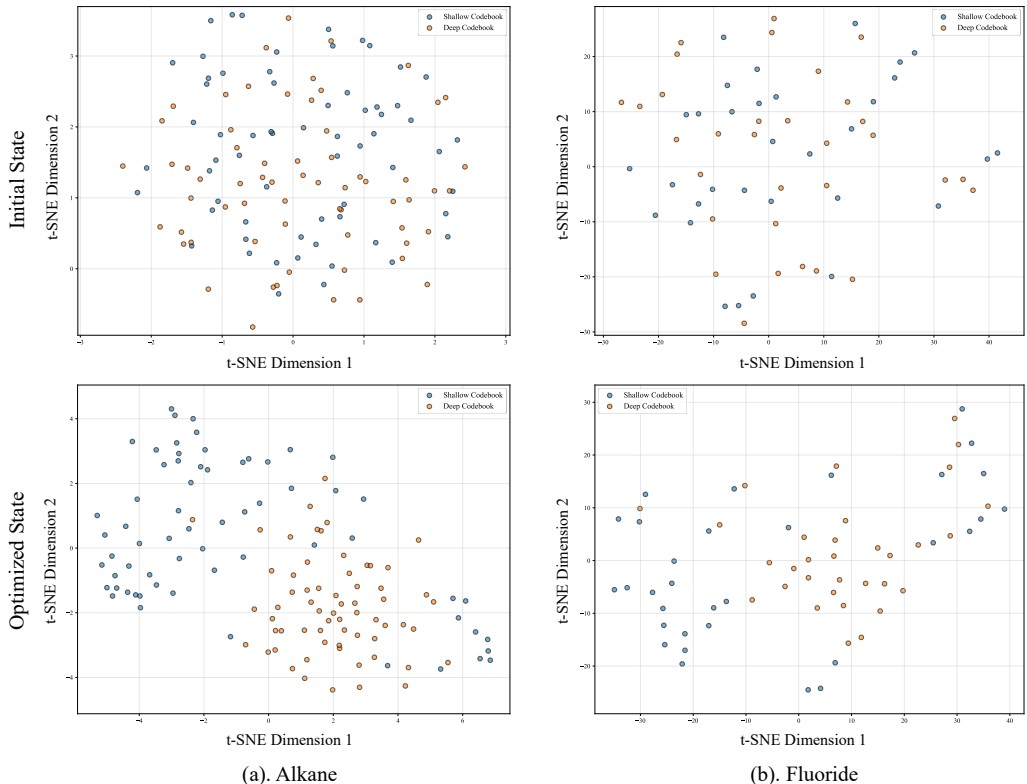

(a). Alkane                    (b). Fluoride

Figure 15: t-SNE visualization of codewords on (a). Alkane dataset and (b) Fluoride dataset.

to the model internal parameters or the model gradients (Pope et al., 2019). Actually, IDEA requires the gray-box accessibility to utilize the GNN encoded representations, which limits the application of IDEA to completely black-box GNN models.

**Approximately linear assumption on unknown mapping function** $\mathcal{H}$. In Appendix G, we introduce a unobserved function $\mathcal{H}$ that maps the latent space $\mathbb{R}^{d'}$ to our prototypical space $\mathbb{R}^d$. The strict validity of $\mathcal{L}_{\text{IDEA}}$ in the explanatory prototype alignment stage necessitates that $\mathcal{H}$ is approximately linear at least. Hence, a highly non-linear function $\mathcal{H}$ might become a potential limitation of IDEA.

## J    FUTURE WORK

A promising direction for future work is to extend the proposed quantization-based explanation framework from instance-level to model-level interpretability. One possibility is to construct a global dictionary of reference quantization prototypes that summarizes the model's decision behavior across the entire dataset. By analyzing how deep quantization patterns cluster in latent space, such a dictionary could reveal class-level structural regularities or decision boundaries, analogous to prototype-based global explanations in prior work. Furthermore, integrating hierarchical or dynamic prototype discovery may help capture more nuanced variations in quantized representations, enabling a more comprehensive characterization of the model's reasoning process.

## K    USE OF LARGE LANGUAGE MODELS

Large language models (LLMs) are used in this work solely for auxiliary purposes. Specifically, they assisted in improving the accuracy of writing by identifying and correcting grammatical issues, refining terminology choices, as well as in suggesting appropriate color schemes for figure design. All research ideas, methodological developments, experiments, and the main body of the manuscript are independently conceived, conducted, and written by the authors.

