# OpenReview forum: "Paradigm Shift of GNN Explainer from Label Space to Prototypical Representation Space"
_ICLR.cc/2026/Conference — ICLR 2026 Poster_

### Official Review · Reviewer_u3DN · 2025-10-27

**Soundness:** 3
**Presentation:** 3
**Contribution:** 3
**Rating:** 6
**Confidence:** 4

**Summary:**

The authors propose a paradigm shift from label space to graph representation space, which enforces the preservation of encoded information when generating local explanations, consequently providing more accurate and robust explanatory subgraphs.

**Strengths:**

- The proposed method is model-agnostic and can be applied to other post-hoc local explanation methods on graphs. Experimental results demonstrate consistent performance improvements across multiple datasets. Moreover, the method exhibits robust performance compared to other baselines under noisy conditions.
- While existing literature has primarily focused on label space, there has been limited research on leveraging informative features from graph representations for explanations. In this regard, this paper presents novelty by introducing a new perspective on graph explainability.
- The paper is well-written and easy to understand.

**Weaknesses:**

- There is a lack of justification for why both shallow and deep quantizers are necessary. While the authors explain that shallow quantization handles non-explanatory substructures and deep quantization captures explanatory substructures, the transition from quantization to disentanglement (similar to MixupExplainer) feels abrupt when first reading the methodology section, which weakens the clarity of presentation. I recommend that the authors provide additional explanation and stronger justification for this design choice.
- The representation in Equations 2 and 3 is a little bit complicated. And there is insufficient explanation of the vector arithmetic operations. The authors are recommended to provide a detailed explanation about equations 2 and 3. Additionally, shouldn't L_D (rather than L_S) in lines 205~7  be the loss that enforces the prediction of the non-explanatory subgraph towards a uniform distribution? (In Figure 2, only y_S is related to the uniform distribution).
- Lack of theoretical analysis.

- Minor issue: in Table 2, rank information is not provided.

**Questions:**

- Why are both shallow and deep quantizers necessary? Is a single quantizer insufficient for your framework? Providing clearer motivation for this dual-quantizer design would significantly enhance the paper's clarity.
- The explanation of vector arithmetic operations in Equations 2 and 3 needs more detail. Could you provide a step-by-step breakdown of these operations?
- Given that you position deep quantization as the relevant source of explanation with reference quantization serving as prototype explanation, could this framework potentially be extended to model-level explanations? Are there promising directions for future work in this area?

Please refer to the weaknesses along with the questions.

---

> ### Author Response · Authors · 2025-11-19
> **Official Comment by Authors (1/2)**
>
> Thanks for your detailed review and insightful comment. Our point-by-point responses are listed as follows.
> > **Weakness 1**. There is a lack of justification for why both shallow and deep quantizers are necessary. While the authors explain that shallow quantization handles non-explanatory substructures and deep quantization captures explanatory substructures, the transition from quantization to disentanglement (similar to MixupExplainer) feels abrupt when first reading the methodology section, which weakens the clarity of presentation. I recommend that the authors provide additional explanation and stronger justification for this design choice.
>
> > **Question 1**. Why are both shallow and deep quantizers necessary? Is a single quantizer insufficient for your framework? Providing clearer motivation for this dual-quantizer design would significantly enhance the paper's clarity.
>
> We apologize for the unclear introduction to the dual-quantizer design. The necessity of both shallow and deep quantizers lies in a key insight in the vector quantization research [1-3]. When conduct vector quantization, the quantizer attempts to approximate the input vector as close as possible. In graph domain, the explanation subgraph tends to be much smaller than the full graph in terms of scale, and thus much smaller than the non-explanation subgraph. Accordingly, within the graph representations, the proportion of non-explanatory information is larger than that of explanatory information. Therefore, if we use a single quantizer design, this quantizer is more likely to encapsulate the non-explanatory information into its codebook.
>
> Regarding the HGTokenizer consisting of two quantizers, the shallow quantizer is used to capture non-explanatory information which contributes the majority of the graph representation, and the deep quantizer aims to capture the explanatory information. This design is consistent with both the characteristics of vector quantization, and the scale difference between the explanatory and non-explanatory subgraphs.

---

> ### Author Response · Authors · 2025-11-19
> **Official Comment by Authors (2/2)**
>
> > **Weakness 2**. The representation in Equations 2 and 3 is a little bit complicated. And there is insufficient explanation of the vector arithmetic operations. The authors are recommended to provide a detailed explanation about equations 2 and 3. Additionally, shouldn't L_D (rather than L_S) in lines 205~7 be the loss that enforces the prediction of the non-explanatory subgraph towards a uniform distribution? (In Figure 2, only y_S is related to the uniform distribution).
>
> > **Question 2**. The explanation of vector arithmetic operations in Equations 2 and 3 needs more detail. Could you provide a step-by-step breakdown of these operations?
>
> We apologize for the confusion in paper reading. Given the input embedding $h_i\in\mathbb R^{1\times d}$, the shallow codebook $\mathcal C_S\in\mathbb R^{K\times d}$, and the deep codebook $\mathcal C_D\in\mathbb R^{K\times d}$, the details of the vector quantization process of Equations (2) and (3) are elaborated as follows.
>
> Step 1. Feed the input embedding $h_i$ into the shallow quantizer. The shallow quantizer first calculates the pair-wise distance between $h_i$ and each codeword $q\in\mathbb R^{1\times d}$ within the shallow codebook $\mathcal C_S$. Afterwards, the shallow quantizer select the closest codeword to $h_i$ according to the $K$-dimensional distance vector. Step 1 is formulated by the formula $q^* _ {S,i}$$={\rm GQ} _ S(h_i)= {\rm argmin}_{q\in \mathcal C_S}\mathcal D(q,h_i)$ in Equation (2).
>
> Step 2. Calculate the quantization residual of the shallow quantizer, which is formulated by the formula $h_i'=h_i-q_{S,i}^*$ in Equation (2).
>
> Step 3. Feed the residual embedding $h_i'$ into the deep quantizer, the detailed quantization process in the same the that in Step 1. The deep quantizer will select the closest codeword $q^* _ {D,i}$$={\rm GQ} _ D(h_i')={\rm argmin}_{q\in \mathcal C_D}\mathcal D(q,h_i')$, as shown in Equation (3).
>
> Subsequently, the quantized representations $q^* _ {S,i}$ and $q^* _ {D,i}$ provide by the shallow and deep quantizers are used to compute the disentanglement loss $\mathcal L_D$. The sum of them, i.e., $q^* _ i=q^* _ {S,i}+q^* _ {D,i}$ in Equation (3), is used to compute the structure-awareness loss $\mathcal L_S$. We have appended the elaboration above in Appendix B.4 Step-by-step Breakdown of HGTokenizer of the revised version.
>
> In addition, regarding the role of $\mathcal L_D$ and $\mathcal L_S$, your understanding is completely correct. $\mathcal L_D$ aims to not only enforce the the prediction of the non-explanatory substructures towards a uniform distribution, but also guide the prediction of the explanatory substructures towards the original prediction. $\mathcal L_S$ aims to enhance the ability of HGTokenizer to capture the graph structural characteristics. The subscripts 'S' and 'D' in prediction $\hat y_S$ and $\hat y_D$ represent 'Shallow' and 'Deep', respectively; while the subscripts 'S' and 'D' in objective $\mathcal L_S$ and $\mathcal L_D$ represent 'Structure-awareness' and 'Disentanglement'.
>
> > **Weakness 4**. Minor issue: in Table 2, rank information is not provided.
>
> Sincerely thanks for your careful reading. We apologize for this writing mistake. The 'Rank' information ought to be 'Improvement' information as defined in the caption of Table 1. We have corrected this typo in the revised version.
>
> > **Question 3**. Given that you position deep quantization as the relevant source of explanation with reference quantization serving as prototype explanation, could this framework potentially be extended to model-level explanations? Are there promising directions for future work in this area?
>
> Your comments are quite far-sighted and we have discussed the future work in Appendix K of the revised version. In fact, there already exist some model-level explanation methods based on prototypes [4-6], while the utilization of vector quantization and codebook remains under-explored. We believe that extending IDAE to model-level explanation is promising, since the codebook can naturally compress large amounts of samples into a compact set of representative centers. This intuition highly matches with the goal of model-level explanation methods.
>
> **Reference**
>
> [1] Additive Quantization for Extreme Vector Compression. CVPR 2014.
>
> [2] Approximate Nearest Neighbor Search by Residual Vector Quantization. Sensors, 2010. (Citation 220)
>
> [3] Adapting Large Language Models by Integrating Collaborative Semantics for Recommendation. ICDE 2024.
>
> [4] Prototype-Based Explanations for Graph Neural Networks. AAAI 2022.
>
> [5] PAGE: Prototype-Based Model-Level Explanations for Graph Neural Networks. IEEE TPAMI 2024.
>
> [6] Global Interpretable Graph-level Anomaly Detection via Prototype. KDD 2025.

---

> > ### Comment · Reviewer_u3DN · 2025-11-27
> > **Response to the authors**
> >
> > My original concerns have been addressed. I maintain my score of 6.

---

> > > ### Author Response · Authors · 2025-11-27
> > >
> > > Thank you sincerely for your comments and suggestions that help us to refine our manuscript.

---

### Official Review · Reviewer_GAVp · 2025-10-27

**Soundness:** 3
**Presentation:** 2
**Contribution:** 3
**Rating:** 6
**Confidence:** 4

**Summary:**

The paper proposes IDEA, a two-stage framework for post-hoc, instance-level GNN explanations. It learns a prototype space of “explanatory” substructures and then aligns the explanation with the original graph in that prototype space using an entropy-regularized objective. Then the paper claimed and positioned as a shift from label-preserving objectives toward representation-level alignment, which aims to better capture fine-grained molecular motifs. After reviewing this paper, I am leaning towards a **weak accept rating (6), since parts of the technical formulation and experimental results, like the transport cost and the disentanglement guarantees, still need necessary clarification.**

**Strengths:**

1. The core pitch, “same label” is often too coarse for chemistry-level reasoning, and aligning explanations in a learned prototype space is a clean and intuitive step forward.

2. The framework is conceptually coherent and is advertised as attachable to different explainer backbones, which increases practical impact.

**Weaknesses:**

1. **Wasserstein cost matrix feels underdefined.** The prototype alignment uses an entropy-regularized Wasserstein distance with cost matrix \(S\). Since the ordering of learned prototypes is arbitrary, for $S_{ij} = (i-j)^2\$ has no obvious semantic meaning. Without a principled metric over prototypes, I am not fully convinced the transport cost is meaningful or stable. Please justify how the indices are ordered or provide an alternative cost.

2. **Granularity / supervision path.** The paper defines distributions $P'_G$ (for the “purified” original graph) and $P_g$ (for the candidate explanation subgraph) over explanatory prototypes, then matches them. It is not fully clear whether these are node-level, edge-level, or graph-level histograms, and how gradients from that distributional alignment are pushed back to edge scores that are later evaluated using ROC-AUC and precision against ground-truth substructures. A short dimensional walkthrough in the main text would make the method easier to trust.

3. **Disentanglement may leak.** The hierarchical tokenizer is trained so that the “shallow” codebook should become non-explanatory and the “deep” codebook should become explanatory, by pushing shallow predictions toward uniform and deep predictions toward the original GNN output. But the same quantized representation is also forced to reconstruct structure (adjacency and node features), which could let shallow codes still absorb predictive motifs.

4. **No std/var or statistical robustness report.** The tables report averages but no standard deviations across random seeds, and there is no statistical test. For some benchmarks the margins over strong baselines are a few percent. Without error bars it is difficult to judge how robust the reported ~4–5% ROC-AUC and ~49% precision improvements really are.

5. **Scope of novelty claim.** The paper repeatedly frames IDEA as a “paradigm shift” from label-preserving explanations to representation-level alignment. I do think the formulation is interesting and practically useful. Still, prior methods like ProxyExplainer already started to address distribution shift between a full graph and an explanatory subgraph using learned proxy distributions, not just raw label matching. So personally, I think the final version should position IDEA as a principled instantiation of this direction, rather than implying absolute first.

6. **(Minor) writing / notation issues.** Some notation is inconsistent or rushed. E.g. the paper alternates between $U^C$, $P_u$, and “uniform” when describing how shallow codes are encouraged to be non-predictive; it is not always obvious these are the same object. There are also a few typos and spots where a symbol appears before it is defined. Cleaning up these points would improve readability.

**Questions:**

1. How is the prototype cost matrix $S_{ij}$ justified? Do you sort prototypes in some interpretable order (e.g., via clustering in latent space) before computing transport cost, or is the index order arbitrary? If it is arbitrary, why does the Wasserstein distance remain meaningful?

2. When defining $P'_G$ and $P_g$, are these distributions aggregated at the graph level, node level, or edge level? How do gradients from the Wasserstein loss translate back into edge selection probabilities in the explainer so that you can evaluate edge-wise ROC-AUC and precision against ground truth substructures? Please clarify the pipeline.

3. Does the shallow codebook actually avoid encoding predictive motifs? Can you quantify how predictive shallow vs deep codebooks are (e.g., accuracy drop if you only use shallow vs only use deep)? Right now LD encourages this split, but LS might leak class-relevant structure back into the shallow part. Some empirical evidence would strengthen the causal interpretation.

4. Can you report mean ± std over multiple seeds for Tables 1–3? Current results seems can not assess robustness of the claimed 4–5% ROC-AUC gain and 40%+ precision gain.

5. For the one negative case (IDEA hurts V-InFoR on Mutag), can you provide evidence on a corrupted-graph benchmark where V-InFoR is useful?

---

> ### Author Response · Authors · 2025-11-19
> **Official Comment by Authors (1/3)**
>
> Thanks for your detailed review and insightful comment. Our point-by-point responses are listed as follows.
> > **Weakness 1**. Wasserstein cost matrix feels underdefined. The prototype alignment uses an entropy-regularized Wasserstein distance with cost matrix (S). Since the ordering of learned prototypes is arbitrary, for  $S_{ij}=(i-j)^2$ has no obvious semantic meaning. Without a principled metric over prototypes, I am not fully convinced the transport cost is meaningful or stable. Please justify how the indices are ordered or provide an alternative cost.
>
> > **Question 1**. How is the prototype cost matrix $S_{ij}$ justified? Do you sort prototypes in some interpretable order (e.g., via clustering in latent space) before computing transport cost, or is the index order arbitrary? If it is arbitrary, why does the Wasserstein distance remain meaningful?
>
> Sincerely thanks for your careful reading and professional comments. We apologize for the incorrect definition of the cost matrix which is a writing mistake. Formally, when calculating the loss $W_\epsilon(X,Y)$, the element $S_{ij}$ in the cost matrix is defined as $(x_i-y_j)^2$, where $x_i$ is the $i$-th element of $X$ and $y_j$ is the $j$-th element of $Y$. Hence, the cost matrix $S$ can measure the cost of aligning the representations of the purified original graph and the explanation subgraph. We have corrected this mistake in the revised version and sorry for our carelessness again.
>
> > **Weakness 2**. Granularity / supervision path. The paper defines distributions $P_G'$ (for the “purified” original graph) and $P_g$ (for the candidate explanation subgraph) over explanatory prototypes, then matches them. It is not fully clear whether these are node-level, edge-level, or graph-level histograms, and how gradients from that distributional alignment are pushed back to edge scores that are later evaluated using ROC-AUC and precision against ground-truth substructures. A short dimensional walkthrough in the main text would make the method easier to trust.
>
> > **Question 2**. When defining $P_G'$ and $P_g$, are these distributions aggregated at the graph level, node level, or edge level? How do gradients from the Wasserstein loss translate back into edge selection probabilities in the explainer so that you can evaluate edge-wise ROC-AUC and precision against ground truth substructures? Please clarify the pipeline.
>
> We apologize for the confusion in reading. First, the two distributions $P_G'$ and $P_g$ are both graph-level histograms, since they are induced by graph-level representations $H_G'$ and $H_g$. To clarify the gradient back-propagation of Wasserstein loss, we elaborate the forward process and backward process of the prototype alignment stage as follows.
>
> Step 1. Feed the input graph $G=(A,X),A\in \mathbb R^{N\times N},X\in\mathbb R^{N\times D}$ into the frozen GNN encoder, and then obtain graph-level representation $H_G\in\mathbb R^{1\times d}$.
>
> Step 2. Feed the input graph $G$ into the learnable explainer backbone (e.g., GNNExplainer or PGExplainer), and the obtain the edge score $\hat e\in\mathbb R^{1\times |E|}$. Edge score $\hat e$ can induce the explanation subgraph $g$ by masking the original edge set $E$. Feed $G$ and $\hat e$ into the frozen GNN encoder and obtain graph-level representation $H_g\in\mathbb R^{1\times d}$ for the explanation subgraph.
>
> Step 3. Following Equation (8) to (10), feed $H_G$ and $H_g$ into the HGTokenizer, then obtain distributions $P_G'\in\mathbb R^{1\times K}$ and $P_g\in\mathbb R^{1\times K}$, where $K$ denotes the number of prototypes.
>
> Step 4. Calculate the Wasserstein loss for $P_G'$ and $P_g$. In our practice, the Wasserstein loss is implemented based on PyTorch and the gradient can be automatically computed by torch.autograd.
>
> Step 5. During the backward process, the gradient back-propagates into the learnable explainer and optimizes the explainer. After optimization, we evaluate the ROC-AUC (precision) based on the ground-truth edge score $e^*\in\{0,1\}^{1\times |E|}$ and the edge score $\hat e\in\mathbb R^{1\times |E|}$ predicted by explainer.

---

> ### Author Response · Authors · 2025-11-19
> **Official Comment by Authors (2/3)**
>
> > **Weakness 3**. Disentanglement may leak. The hierarchical tokenizer is trained so that the “shallow” codebook should become non-explanatory and the “deep” codebook should become explanatory, by pushing shallow predictions toward uniform and deep predictions toward the original GNN output. But the same quantized representation is also forced to reconstruct structure (adjacency and node features), which could let shallow codes still absorb predictive motifs.
>
> > **Question 3**. Does the shallow codebook actually avoid encoding predictive motifs? Can you quantify how predictive shallow vs deep codebooks are (e.g., accuracy drop if you only use shallow vs only use deep)? Right now LD encourages this split, but LS might leak class-relevant structure back into the shallow part. Some empirical evidence would strengthen the causal interpretation.
>
> Thanks for your insightful comments. First, it is impossible for the disentanglement stage to completely eliminate the explanatory information from the shallow codebook. However, within IDEA, the deep codebook can encapsulate the majority of the explanatory information. Following your instruction, we have conducted new experiment to evaluate the prediction accuracy of the shallow quantized representation and the deep quantized representation, respectively, to investigate how predictive they are. For each dataset, the GNN model in this experiment is the same one as that to be explained. The evaluated results are presented as follows.
>
> Dataset|Mutagenicity|Benzene|Alkane|Fluoride|BA-2Motifs
> -|-|-|-|-|-
> Original|0.8300|0.9054|0.9620|0.9340|1.00
> Shallow|0.5463|0.6175|0.6472|0.4497|0.5550
> Deep|0.6725|0.8250|0.9080|0.8342|0.8955
>
> According to the results above, one can see that there exists a significant predictive gap between the shallow codebook and the deep codebook. For Mutagenicity, Fluoride, and BA-2Motifs, the predictive ability of shallow codebook is close to random prediction. For Benzene and Alkane, though the accuracy of shallow codebooks exceeds 0.60, the much higher accuracy of deep codebooks (0.8250 and 0.9080) indicate that majority of explanatory information is captured.
>
> > **Weakness 4**. No std/var or statistical robustness report. The tables report averages but no standard deviations across random seeds, and there is no statistical test. For some benchmarks the margins over strong baselines are a few percent. Without error bars it is difficult to judge how robust the reported ~4–5% ROC-AUC and ~49% precision improvements really are.
>
> > **Question 4**. Can you report mean ± std over multiple seeds for Tables 1–3? Current results seems can not assess robustness of the claimed 4–5% ROC-AUC gain and 40%+ precision gain.
>
> We apologize for the missing of standard deviations and statistical test. We have updated Tables 1-3 in the revised version. The results demonstrate that, in the vast majority, the improvements of IDEA are statistically significant with $p$-value less than 0.01. The results are averaged over ten different random seeds. All the models in our experiment are fairly compared with the same dataset split and random seed set.
>
> > **Weakness 5.** Scope of novelty claim. The paper repeatedly frames IDEA as a “paradigm shift” from label-preserving explanations to representation-level alignment. I do think the formulation is interesting and practically useful. Still, prior methods like ProxyExplainer already started to address distribution shift between a full graph and an explanatory subgraph using learned proxy distributions, not just raw label matching. So personally, I think the final version should position IDEA as a principled instantiation of this direction, rather than implying absolute first.
>
> Thanks for your insightful suggestion. We agree that prior works like ProxyExplainer have push forward to address distribution shift through learned proxy distributions. In the final version, we will not claim to be the absolute first in moving beyond label-preserving explanations, and further elaborate the fundamental contribution of IDEA lies in the representation-level alignment.
>
> > **Weakness 6**. (Minor) writing / notation issues. Some notation is inconsistent or rushed. E.g. the paper alternates between $U^C$, $P_u$, and “uniform” when describing how shallow codes are encouraged to be non-predictive; it is not always obvious these are the same object. There are also a few typos and spots where a symbol appears before it is defined. Cleaning up these points would improve readability.
>
> We apologize for the writing mistakes, including the uniform distribution as you point out and the Wasserstein cost matrix $S$ (in Weakness 1). In the revised version, we have double checked the notation definition and paper writing, and corrected the typos and inconsistent notations.

---

> ### Author Response · Authors · 2025-11-19
> **Official Comment by Authors (3/3)**
>
> > **Question 5**. For the one negative case (IDEA hurts V-InFoR on Mutag), can you provide evidence on a corrupted-graph benchmark where V-InFoR is useful?
>
> Following your insightful comment, we introduce random noise into Mutagenicity and obtain structurally corrupted datasets, as the evaluation experiment in V-InFoR. Specifically, we evaluate IDAE with four explainer backbones (same to Table 3) on Mutagenicity with 15% and 30% noise. Since the ground-truth explanation is meaningless after introducing noise, we adopt the faithfulness metrics to measure the explanation performance. Fidelity$ _ +$ measures the change degree of the GNN prediction after removing the explanation subgraph, Fidelity$ _ -$ measures the change degree of the GNN prediction when only retain the explanation subgraph. For readability, we report Fidelity$ _ +$, 1-Fidelity$ _ -$, and their harmonic mean, which are better when they are higher and belong to $[0,1]$. The results are presented as follows.
>
> Fidelity$_+$|w/o Noise|0.15 Noise|0.30 Noise
> -|-|-|-
> PGExplainer|$0.2012_{\pm 0.0097}$|$0.5360_{\pm  0.0029 }$|$0.6647_{\pm  0.0029 }$
> +IDEA|$0.2207_{\pm  0.0093 }$|$0.5693_{\pm  0.0068 }$|$0.6982_{\pm  0.0004 }$
> *Improvement*|$9.69\\%$|$6.21\\%$|$5.04\\%$|
> ReFine|$0.2161_{\pm  0.0041 }$|$0.4983_{\pm  0.0075 }$|$0.6391_{\pm  0.0027 }$
> +IDEA|$0.3254_{\pm  0.0096 }$|$0.5156_{\pm  0.0020 }$|$0.6774_{\pm  0.0110 }$
> *Improvement*|$50.58\\%$|$3.47\\%$|$5.99\\%$|
> V-InFoR|$0.1954_{\pm  0.0004 }$|$0.7485_{\pm  0.0057 }$|$0.4825_{\pm  0.0079 }$
> +IDEA|$0.1957_{\pm  0.0006 }$|$0.7732_{\pm  0.0006 }$|$0.5395_{\pm  0.0069 }$
> *Improvement*|$0.15\\%$|$3.30\\%$|$11.81\\%$|
> ProxyExplainer|$0.1841_{\pm  0.0132 }$|$0.1307_{\pm  0.0006 }$|$0.1997_{\pm  0.0028 }$
> +IDEA|$0.1992_{\pm  0.0129 }$|$0.1515_{\pm  0.0032 }$|$0.2118_{\pm  0.0048 }$
> *Improvement*|$8.2\\%$|$15.91\\%$|$6.06\\%$|
>
> 1-Fidelity$_-$|w/o Noise|0.15 Noise|0.30 Noise
> -|-|-|-
> PGExplainer|$0.7714_{\pm 0.0165}$|$0.4324_{\pm  0.0085 }$|$0.2554_{\pm  0.0125 }$
> +IDEA|$0.8018_{\pm  0.0086 }$|$0.4688_{\pm  0.001 0}$|$0.3153_{\pm  0.0013 }$
> *Improvement*|$3.94\\%$|$8.42\\%$|$23.45\\%$|
> ReFine|$0.6604_{\pm  0.0044 }$|$0.4536_{\pm  0.0069 }$|$0.3336_{\pm  0.0027 }$
> +IDEA|$0.6665_{\pm  0.0069 }$|$0.4738_{\pm  0.003 0}$|$0.3508_{\pm  0.0101 }$
> *Improvement*|$0.92\\%$|$4.45\\%$|$5.16\\%$|
> V-InFoR|$0.6375_{\pm  0.0020}$|$0.4825_{\pm  0.0067 }$|$0.3963_{\pm  0.0028 }$
> +IDEA|$0.6373_{\pm  0.0057 }$|$0.5395_{\pm  0.0023 }$|$0.4174_{\pm  0.0089 }$
> *Improvement*|$-0.03\\%$|$11.81\\%$|$5.32\\%$|
> ProxyExplainer|$0.7912_{\pm  0.0058 }$|$0.8205_{\pm  0.0027 }$|$0.8009_{\pm  0.0077 }$
> +IDEA|$0.9180_{\pm  0.0032 }$|$0.8513_{\pm  0.0005 }$|$0.8099_{\pm  0.0051 }$
> *Improvement*|$16.03\\%$|$3.75\\%$|$1.12\\%$|
>
> Harmonic Mean|w/o Noise|0.15 Noise|0.30 Noise
> -|-|-|-
> PGExplainer|$0.3191_{\pm 0.0136}$|$0.4786 _{\pm  0.0063 }$|$0.3689 _{\pm  0.0131 }$
> +IDEA|${0.3460} _{\pm  0.0114 }$|$0.5142 _{\pm  0.0033 }$|$0.4344 _{\pm  0.0013 }$
> *Improvement*|$8.43\\%$|$7.44\\%$|$17.76\\%$|
> ReFine|${0.3256} _{\pm  0.0051 }$|$0.4749 _{\pm  0.0070 }$|$0.4384 _{\pm  0.0030 }$
> +IDEA|$0.4373 _{\pm  0.0100 }$|$0.4938 _{\pm  0.0025 }$|$0.4622 _{\pm  0.0111 }$
> *Improvement*|$34.34\\%$|$3.98\\%$|$5.43\\%$|
> V-InFoR|$0.2991 _{\pm  0.0006 }$|$0.5867 _{\pm  0.0064 }$|$0.4352 _{\pm  0.0048 }$
> +IDEA|$0.2994 _{\pm  0.0014 }$|$0.6355 _{\pm  0.0018 }$|$0.4706 _{\pm  0.0082 }$
> *Improvement*|$0.1\\%$|$8.32\\%$|$8.13\\%$|
> ProxyExplainer|$0.2985 _{\pm  0.0178 }$|$0.2255 _{\pm  0.0011 }$|$0.3197 _{\pm  0.0041 }$
> +IDEA|$0.3272 _{\pm  0.0176 }$|$0.2572 _{\pm  0.0046 }$|$0.3358 _{\pm  0.0064 }$
> *Improvement*|$9.61\\%$|$14.06\\%$|$5.04\\%$|
>
> According to the results above, one can have the following two observations. First, V-InFoR exhibits robust explanation performance on corrupted graphs. Second, when explain corrupted graphs, combining V-InFoR with IDEA can further improvement the explanation performance.

---

> > ### Author Response · Authors · 2025-11-28
> >
> > Dear Reviewer GAVp,
> >
> > We sincerely appreciate your valuable comments and thoughtful suggestions. Following your advice, we have provided additional clarifications and further experiments. We hope these efforts address your concerns effectively.
> >
> > Since the end of the discussion is approaching, we sincerely look forward to your reevaluation and would be deeply grateful if you could consider a higher rating.
> >
> > Best regards,
> > Authors of paper 858

---

### Official Review · Reviewer_dsYU · 2025-11-01

**Soundness:** 3
**Presentation:** 3
**Contribution:** 2
**Rating:** 4
**Confidence:** 3

**Summary:**

This paper identifies a limitation in existing post-hoc GNN explainers: they are typically optimized under a label-preserving framework, which aligns the GNN's label prediction for the original graph with that of the explanation subgraph. The authors argue that this label space is not expressive enough to capture the rich structural information GNNs use, especially when multiple distinct graph structures can map to the same label. The authors first demonstrate that a naive Direct Alignment in this space is ineffective due to two key challenges and provide a novel dual-stage optimization framework designed to overcome these challenges. Experiments show that IDEA outperforms state-of-the-art baselines.

**Strengths:**

S1. The paper clearly and convincingly articulates a fundamental, yet often-overlooked, weakness of the dominant label-preserving framework.

S2. The proposed IDEA framework, with its use of vector quantization for disentanglement and prototypical distribution alignment, provides a reasonable solution to both problems.

S3. IDEA demonstrates consistent and significant performance gains over a wide range of SOTA eplainers on five datasets

**Weaknesses:**

W1. The entire framework's success hinges on the Stage 1 HGTokenizer correctly separating explanatory from non-explanatory information. This separation is enforced by the proxy objective $\mathcal{L}_D$, which pushes the $Q_S$-based prediction to a uniform distribution and the $Q_D$-based prediction to the original GNN prediction. While this is a common proxy, it's not a guarantee. It's plausible that in some cases, the explanatory signal could be simple and captured by the shallow quantizer, while a complex confounder is relegated to the deep quantizer.

W2. The theoretical justification in Appendix G for using assignment probability as a proxy for latent space location is insightful but relies on several strong, unverified assumptions. Specifically, it assumes that the (unobserved) mapping from the true latent space to the prototypical space is linear or can be approximated linearly. It is not clear how the method would perform if this mapping is highly non-linear, which is very possible in deep models. A discussion of this limitation or a more robust justification would be beneficial.

W3. The paper relies on ground-truth-based metrics (ROC-AUC and Precision). While standard, GNN explanation evaluation also benefits from model-centric metrics that measure faithfulness (i.e., how much the model's prediction actually changes when the identified subgraph is removed or masked).

**Questions:**

Please refer to the weaknesses above.

---

> ### Author Response · Authors · 2025-11-19
> **Official Comment by Authors (1/3)**
>
> Thanks for your detailed review and insightful comment. Our point-by-point responses are listed as follows.
> > **Weakness 1**. The entire framework's success hinges on the Stage 1 HGTokenizer correctly separating explanatory from non-explanatory information. This separation is enforced by the proxy objective $\mathcal L_D$, which pushes the $Q_S$-based prediction to a uniform distribution and the $Q_D$-based prediction to the original GNN prediction. While this is a common proxy, it's not a guarantee. It's plausible that in some cases, the explanatory signal could be simple and captured by the shallow quantizer, while a complex confounder is relegated to the deep quantizer.
>
> Within the HGTokenizer, we design the objective $\mathcal L_D$ for the shallow and deep quantizer based on a key insight in the vector quantization research [1-3]. When conduct hierarchical vector quantization, the $l$-th quantizer takes the quantization residual of the $(l-1)$-th quantizer. Since each quantizer attempts to approximate the input vector as close as possible, the element values of shallow quantized vectors are usually larger than those of the deep quantized vectors components. In graph domain, the explanation subgraph tends to be much smaller than the full graph in terms of scale, and thus much smaller than the non-explanation subgraph. Accordingly, within the graph representations, the proportion of non-explanatory information is larger than that of explanatory information. Therefore, in objective $\mathcal L_D$, the shallow quantizer is used to capture non-explanatory information which contributes the majority of the graph representation, and the deep quantizer aims to capture the explanatory information. This design is consistent with both the residual characteristics of hierarchical quantization, and the scale difference between the explanatory and non-explanatory subgraphs.
>
> During practice, we have attempted to exchange the role of shallow and deep quantizers, i.e., we use the shallow quantizer to capture explanatory substructures and the deep quantizer to capture non-explanatory ones. The explanation performance comparison of IDEA and this inverse variant is presented as follows.
> ROC-AUC|Mutagenicity|Benzene|Alkane|Fluoride|BA-2Motifs
> -|-|-|-|-|-
> IDAE|$0.7379$|$0.9138$|$0.9355$|$0.8868$|$0.9541$
> Inverse IDEA|$0.6549$|$0.8983$|$0.8972$|$0.7476$|$0.8459$
>
> According to the results, IDEA can consistently outperform the inverse variant, which can empirically support the reasonability of the HGTokenizer design.
>
> > **Weakness 2**.  The theoretical justification in Appendix G for using assignment probability as a proxy for latent space location is insightful but relies on several strong, unverified assumptions. Specifically, it assumes that the (unobserved) mapping from the true latent space to the prototypical space is linear or can be approximated linearly. It is not clear how the method would perform if this mapping is highly non-linear, which is very possible in deep models. A discussion of this limitation or a more robust justification would be beneficial.
>
> First, following your valuable comment, we have appended related discussion about the highly non-linear latent mapping function in the Appendix J of revised version, which is a potential limitation of IDEA effectiveness. Second, the latent mapping function $\mathcal H$ here represents an objectively existing relationship between the unobserved decisive factors and the learned prototypes, which does not depend on any deep learning models. Empirically, according to the explanation performance of IDEA on the evaluated datasets, the approximately linear assumption of $\mathcal H$ is feasible in practice.

---

> ### Author Response · Authors · 2025-11-19
> **Official Comment by Authors (2/3)**
>
> > **Weakness 3**. The paper relies on ground-truth-based metrics (ROC-AUC and Precision). While standard, GNN explanation evaluation also benefits from model-centric metrics that measure faithfulness (i.e., how much the model's prediction actually changes when the identified subgraph is removed or masked).
>
> Following your valuable suggestion, we report the faithfulness metrics in the revised version, including Fidelity$ _ +$, Fidelity$ _ -$. Fidelity$ _ +$ measures the change degree of the GNN prediction after removing the explanation subgraph, while Fidelity$ _ -$ measures the change degree of the GNN prediction when only retain the explanation subgraph. For readability, we report Fidelity$ _ +$, 1-Fidelity$ _ -$, and their harmonic mean, which are better when they are higher and belong to $[0,1]$.
>
> For convenience, the faithfulness metrics are transcribed below. The results also demonstrate the superiority of IDEA when compared with the SOTA baselines.
>
> Fidelity$_+$|Mutagenicity|Benzene|Alkane|Fluoride|BA-2Motifs
> -|-|-|-|-|-
> GNNExplainer|$0.2136_{\pm 0.0005}$|$0.5614_{\pm 0.0005}$|$0.5435_{\pm 0.0130}$|$0.1242_{\pm 0.0026}$|$0.4067_{\pm 0.0033}$
> PGExplainer|$0.2012_{\pm 0.0097}$|$0.7250_{\pm 0.0028}$|$\underline{0.7826}_{\pm 0.0063}$|$0.4097_{\pm 0.0118}$|$\underline{0.4375}_{\pm 0.0030}$
> GraphMask|$0.0982_{\pm 0.0080}$|$0.4450_{\pm 0.0169}$|$0.5659_{\pm 0.0180}$|$0.2070_{\pm  0.0029 }$|$0.3750_{\pm  0.0043 }$
> ReFine|$\underline{0.2161}_{\pm  0.0041 }$|$0.5690_{\pm  0.0048 }$|$0.6224_{\pm  0.0033 }$|$0.6132_{\pm  0.0077 }$|$0.2068_{\pm  0.0024 }$
> V-InFoR|$0.1954_{\pm  0.0004 }$|$0.5265_{\pm  0.0031 }$|$0.6883_{\pm  0.0013 }$|$0.6298_{\pm  0.0005 }$|$0.3793_{\pm  0.0058 }$
> D4Explainer|$0.0698_{\pm  0.0181 }$|$0.5248_{\pm  0.008 0}$|$0.6093_{\pm  0.0058 }$|$0.6047_{\pm  0.0026 }$|$0.2127_{\pm  0.0014 }$
> MixupExplainer|$0.1277_{\pm  0.0074 }$|$0.4910_{\pm  0.0047 }$|$0.4579_{\pm  0.0053 }$|$0.3672_{\pm  0.0009 }$|$0.2131_{\pm  0.0082 }$
> ProxyExplainer|$0.1841_{\pm  0.0132 }$|$\underline{0.7473}_{\pm  0.0118 }$|$0.6904_{\pm  0.0052 }$|$\underline{0.6607}_{\pm  0.0351 }$|$0.3064_{\pm  0.0027 }$
> IDEA|$\mathbf{0.2207}_{\pm  0.0093 }$|$\mathbf{0.8292}_{\pm  0.0081 }$|$\mathbf{0.8043}_{\pm  0.016 0}$|$\mathbf{0.6988}_{\pm  0.0042 }$|$\mathbf{0.4450}_{\pm  0.0004 }$
> *Improvement*|$2.13\\%$|$10.96\\%$|$2.77\\%$|$5.77\\%$|$1.71\\%$
>
> 1-Fidelity$_-$|Mutagenicity|Benzene|Alkane|Fluoride|BA-2Motifs
> -|-|-|-|-|-
> GNNExplainer|$0.5975_{\pm 0.0053}$|$0.4370_{\pm 0.0051}$|$0.1658_{\pm 0.0070}$|$0.2679_{\pm 0.0114}$|$0.7366_{\pm 0.0031}$
> PGExplainer|$0.7714_{\pm 0.0165}$|$0.5222_{\pm 0.0037}$|$0.3787_{\pm 0.0048}$|$0.2232_{\pm 0.0167}$|$0.9055_{\pm 0.0033}$
> GraphMask|$0.6174_{\pm 0.0032}$|$0.4365_{\pm 0.0065}$|$0.2683_{\pm 0.0058}$|$0.1487_{\pm  0.0013 }$|$0.5005_{\pm  0.0051 }$
> ReFine|$0.6604_{\pm  0.0044 }$|$0.5056_{\pm  0.0082 }$|$0.3237_{\pm  0.0022 }$|$0.2863_{\pm  0.0104 }$|$0.8330_{\pm  0.014 }$
> V-InFoR|$0.6375_{\pm  0.0020}$|$0.4524_{\pm  0.0039 }$|$0.3886_{\pm  0.007 0}$|$0.2871_{\pm  0.0004 }$|$0.7872_{\pm  0.0093 }$
> D4Explainer|$0.6451_{\pm  0.024 }$|$0.4497_{\pm  0.0019 }$|$0.3691_{\pm  0.0086 }$|$0.2577_{\pm  0.0151 }$|$0.9710_{\pm  0.0038 }$
> MixupExplainer|$0.6745_{\pm  0.0115 }$|$0.4962_{\pm  0.0063 }$|$0.3750_{\pm  0.0014 }$|$0.2665_{\pm  0.0051 }$|$0.9513_{\pm  0.0077 }$
> ProxyExplainer|$\underline{0.7912}_{\pm  0.0058 }$|$\underline{0.6483}_{\pm  0.0156 }$|$\mathbf{0.4191}_{\pm  0.0028 }$|$\underline{0.3594}_{\pm  0.0177 }$|$\underline{0.9697}_{\pm  0.0062 }$
> IDEA|$\mathbf{0.8018}_{\pm  0.0086 }$|$\mathbf{0.6964}_{\pm  0.0148 }$|$\underline{0.4190}_{\pm  0.0158 }$|$\mathbf{0.3612}_{\pm  0.001 0}$|$\mathbf{0.9981}_{\pm  0.0003 }$
> *Improvement*|$1.34\\%$|$7.42\\%$|-$0.02\\%$|$5.00\\%$|$2.93\\%$

---

> ### Author Response · Authors · 2025-11-19
> **Official Comment by Authors (3/3)**
>
> Harmonic Mean|Mutagenicity|Benzene|Alkane|Fluoride|BA-2Motifs
> -|-|-|-|-|-
> GNNExplainer|$0.3146_{\pm 0.0013}$|$0.4914_{\pm 0.0031}$|$0.2541_{\pm 0.0096}$|$0.1696_{\pm 0.0019}$|$0.5240_{\pm 0.0035}$
> PGExplainer|$0.3191_{\pm 0.0136}$|$0.6071_{\pm 0.0035}$|$0.5104_{\pm 0.0056}$|$0.2888_{\pm 0.0169}$|$\underline{0.5899}_{\pm 0.0034}$
> GraphMask|$0.1693_{\pm 0.0119}$|$0.4404_{\pm 0.0069}$|$0.3640_{\pm 0.0089}$|$0.1731_{\pm 0.0019}$|$0.4287 _{\pm  0.0031 }$
> ReFine|$\underline{0.3256} _{\pm  0.0051 }$|$0.5354 _{\pm  0.0066 }$|$0.4259 _{\pm  0.0027 }$|$0.3903 _{\pm  0.011 }$|$0.3313 _{\pm  0.0036 }$
> V-InFoR|$0.2991 _{\pm  0.0006 }$|$0.4866 _{\pm  0.0035 }$|$0.4967 _{\pm  0.0060 }$|$0.3944 _{\pm  0.0005 }$|$0.5119 _{\pm  0.0063 }$
> D4Explainer|$0.1254 _{\pm  0.0301 }$|$0.4843 _{\pm  0.0042 }$|$0.4597 _{\pm  0.0080 }$|$0.3612 _{\pm  0.0152 }$|$0.3490 _{\pm  0.0021 }$
> MixupExplainer|$0.2146 _{\pm  0.0106 }$|$0.4935 _{\pm  0.0034 }$|$0.4123 _{\pm  0.0030 }$|$0.3088 _{\pm  0.0037 }$|$0.3481 _{\pm  0.0106 }$
> ProxyExplainer|$0.2985 _{\pm  0.0178 }$|$\underline{0.6943 }_{\pm  0.014 }$|$\underline{0.5216} _{\pm  0.0036 }$|$\underline{0.4655} _{\pm  0.0232 }$|$0.4657 _{\pm  0.0036 }$
> IDEA|$\mathbf{0.3460} _{\pm  0.0114 }$|$\mathbf{0.7569} _{\pm  0.0119 }$|$\mathbf{0.5509} _{\pm  0.0171 }$|$\mathbf{0.4762} _{\pm  0.0018 }$|$\mathbf{0.6156}_{\pm  0.0004 }$
> *Improvement*|$6.26\\%$|$9.02\\%$|$5.62\\%$|$2.30\\%$|$4.36\\%$
>
> **Reference**
>
> [1] Additive Quantization for Extreme Vector Compression. CVPR 2014.
>
> [2] Approximate Nearest Neighbor Search by Residual Vector Quantization. Sensors, 2010. (Citation 220)
>
> [3] Adapting Large Language Models by Integrating Collaborative Semantics for Recommendation. ICDE 2024.

---

> ### Author Response · Authors · 2025-11-28
>
> Dear Reviewer dsYU,
>
> We sincerely appreciate your valuable comments and thoughtful suggestions. Following your advice, we have provided additional clarifications and further experiments. We hope these efforts address your concerns effectively.
>
> Since the end of the discussion is approaching, we sincerely look forward to your reevaluation and would be deeply grateful if you could consider a higher rating.
>
> Best regards,
> Authors of paper 858

---

### Official Review · Reviewer_dWjn · 2025-11-01

**Soundness:** 3
**Presentation:** 3
**Contribution:** 3
**Rating:** 4
**Confidence:** 4

**Summary:**

This paper proposes a novel paradigm shift for post-hoc instance-level GNN explanation: moving the optimization target from the label space to a learned prototypical representation space. The authors identify two key obstacles in directly aligning GNN-encoded representations: (1) entanglement of explanatory and non-explanatory substructures, and (2) distributional mismatch between the original input graph and its sparse explanation subgraph.

To tackle these, the authors introduce IDEA, a two-stage optimization framework. In the first stage, a Hierarchical Graph Tokenizer (HGTokenizer) equipped with a structure-aware disentanglement (SAD) objective separates explanatory from non-explanatory components and clusters the explanatory parts into prototypes. In the second stage, assignment distributions of the purified input and explanation subgraphs over these prototypes are aligned via an entropy-regularized Wasserstein distance.

Extensive experiments across both real-world and synthetic datasets demonstrate IDEA's superiority in ROC-AUC and precision, and its generalizability across various explainer backbones is also validated.

**Strengths:**

1. The shift from label alignment to prototype-based representation alignment is novel and addresses a fundamental limitation in current GNN explainers.

2. The two-stage IDEA framework is carefully designed. The use of shallow and deep vector quantization for disentangling, and Wasserstein distance for alignment, is both elegant and sound.

3. Comprehensive experiments across diverse datasets and explainer architectures convincingly validate the effectiveness and generality of IDEA. Ablation studies, visualization, and extended appendix analyses (e.g., noise robustness, hyperparameter analysis) are particularly thorough.

4. The paper is exceptionally well-written, and all figures and diagrams are highly informative and polished.

**Weaknesses:**

1. **Limited dataset diversity**: The method is only evaluated on five datasets. Testing on more diverse or larger real-world datasets (e.g., citation networks, program graphs) would strengthen the paper’s applicability.

2. **Lack of clarity on prototype initialization**: The paper does not describe how the prototype embeddings (codewords in $C_S$ and $C_D$) are initialized. This is important for reproducibility and could affect convergence behavior.

3. **Possible invalid supervision in Equation (5)**: The second term of Equation (5) uses CrossEntropy between $\hat{y}_D$ and $\hat{y}$. However, $\hat{y}_D$ is generated from the deep quantized representations $q^\*_D$, which are OOD with respect to the frozen GNN predictor. If the predictor was trained on original (non-disentangled) representations, its output for $q^\*_D$ may not be meaningful.

4. **Residual OOD inconsistency in optimization**: Although the paper claims that aligning in representation space avoids the OOD issue, Equation (5) still uses frozen GNN predictions $\hat{y}$ as supervision for deep quantized vectors $\hat{y}_D$, which remain out-of-distribution. This inconsistency needs to be addressed.

5. **Incomplete related work**: The paper largely overlooks existing prototype-based explanation approaches. In addition to Dai & Wang (2025), key missing references include:
   - ProtGNN (Zhang et al., AAAI 2021)
   - PAGE (Shin et al., TPAMI 2022)
   - Prototype-Based Interpretable GNNs (Ragno et al., IEEE TAI 2024)
   - Towards Prototype-Based Self-Explainable GNN (Dai & Wang, TKDD 2022)

**Questions:**

1. Regarding Equation (5): Could you justify why minimizing CrossEntropy($\hat{y}_D$, $\hat{y}$) is valid given that $\hat{y}_D$ is OOD with respect to the frozen predictor?
2. Could you evaluate IDEA on larger or structurally more complex datasets to support scalability?
3. Is it always appropriate to enforce the shallow quantized representation to produce a uniform prediction distribution (first term in Equation (5))?
4. How are the prototype codebooks $C_S$ and $C_D$ initialized? Is the method sensitive to different initialization schemes?
5. Can the authors provide qualitative or quantitative analysis of the learned prototypes? E.g., do deep prototypes align with known substructures (e.g., benzene rings)?
6. Do shallow and deep prototypes exhibit distinguishable structural roles? Can visualizations or clustering be used to support this?
7. Some baseline results seem lower than in original papers. Are all results averaged over multiple runs? Could standard deviations be reported?
8. Hyperparameter tuning varies significantly across datasets (e.g., codebook size $K$). How practical is IDEA in scenarios without access to per-dataset tuning resources?
9. In Table 3, IDEA with V-InFoR shows performance degradation on Mutagenicity. Can the authors explain why IDEA is incompatible in this case?
10. The paper mentions that node-level tasks can be reformulated as graph-level tasks using computation graphs. Could the authors validate this claim with experiments, or clarify the scope as graph-level only?
11. How does IDEA differ fundamentally from prior prototype-based GNN explainers? Is the key contribution the new optimization paradigm (label space → representation space), or the hierarchical disentanglement mechanism?

---

> ### Author Response · Authors · 2025-11-19
> **Official Comment by Authors (1/4)**
>
> Thanks for your detailed review and insightful comment. Our point-by-point responses are listed as follows.
> > **Weakness 1**. Limited dataset diversity: The method is only evaluated on five datasets. Testing on more diverse or larger real-world datasets (e.g., citation networks, program graphs) would strengthen the paper’s applicability.
>
> > **Question 2**. Could you evaluate IDEA on larger or structurally more complex datasets to support scalability?
>
> Following your valuable comment, we further evaluate IDEA on three larger real-world node classification datasets, including two citation networks (Cora and PubMed) and one e-commerce network (Amazon-Computers). The dataset statistics are summarized below.
>
> Dataset|#Nodes|#Edges|#Features|#Classes
> -|-|-|-|-
> Cora|2,708|10,556|1,433|7
> PubMed|19,717|88,648|3,703|6
> Computers|13,752|491,722|7,67|10
>
> Since these real-world datasets lack ground-truth explanation, we adopt widely-used Fidelity$ _ +$ and Fidelity$ _ -$ as the evaluated metrics following standard evaluation [1]. Fidelity$ _ +$ measures the change degree of the GNN prediction after removing the explanation subgraph, while Fidelity$ _ -$ measures the change degree of the GNN prediction when only retain the explanation subgraph. For readability, we report Fidelity$ _ +$, 1-Fidelity$ _ -$, and their harmonic mean, which are better when they are higher and belong to $[0,1]$. Since some novel baselines in the main experiment do not explicitly support node classification datasets, we append three classic explanation methods Saliency, Integrated Gradient (IG), and Guided Backpropagation (GuidedBP) into the evaluated baselines.
>
> The evaluation results on three datasets are presented in the following three tables.
>
> Cora|Fidelity$_+$|1-Fidelity$_-$|Harmonic-Mean
> -|-|-|-
> Saliency|$0.1134$|$0.2660$|$0.1590$
> IG|$0.1754$|$0.4966$|$\underline{0.2592}$
> GuidedBP|$\underline{0.2080}$|$0.3178$|$0.2514$
> GNNExplainer|$0.0814_{\pm0.0026}$|$\underline{0.9595}_{\pm 0.0012}$|$0.1501_{\pm 0.0044}$
> PGExplainer|$0.0875_{\pm 0.0097}$|$0.9258_{\pm 0.0049}$|$0.1597_{\pm 0.0162}$
> GraphMask|$0.0589_{\pm 0.0038}$|$0.9374_{\pm 0.0027}$|$0.1108_{\pm 0.0067}$
> IDEA|$\mathbf{0.2893}_{\pm 0.0045}$|$\mathbf{0.9854}_{\pm 0.0007}$|$\mathbf{0.4472}_{\pm 0.0055}$
> *Improvement*|$39.09\\%$|$2.70\\%$|$72.53\\%$
>
> PubMed|Fidelity$_+$|1-Fidelity$_-$|Harmonic-Mean
> -|-|-|-
> Saliency|$0.1354$|$0.5782$|$0.2194$
> IG|$0.0770$|$0.5392$|$0.1348$
> GuidedBP|$0.0735$|$0.6689$|$0.1324$
> GNNExplainer|$\underline{0.1401}_{\pm 0.0087}$|$0.8340_{\pm 0.0029}$|$\underline{0.2398}_{\pm 0.0127}$
> PGExplainer|$0.0943_{\pm 0.0086}$|$0.7615_{\pm 0.0200}$|$0.1677_{\pm 0.0141}$
> GraphMask|$0.1099_{\pm 0.0004}$|$\underline{0.8895}_{\pm 0.0027}$|$0.1956_{\pm 0.0007}$
> IDEA|$\mathbf{0.1489}_{\pm 0.0003}$|$\mathbf{0.9473}_{\pm 0.0084}$|$\mathbf{0.2573}_{\pm 0.0007}$
> *Improvement*|$6.28\\%$|$13.15\\%$|$7.30\\%$
>
> Computers|Fidelity$_+$|1-Fidelity$_-$|Harmonic-Mean
> -|-|-|-
> Saliency|$0.0233$|$0.3664$|$0.0438$
> IG|$0.0531$|$0.3828$|$0.0932$
> GuidedBP|$0.0629$|$0.3711$|$0.1075$
> GNNExplainer|$0.4798_{\pm 0.0102}$|$0.5736_{\pm 0.0128}$|$0.5225_{\pm 0.0111}$
> PGExplainer|$\underline{0.5972}_{\pm 0.0023}$|$\underline{0.8495}_{\pm 0.0209}$|$\underline{0.7013}_{\pm 0.0086}$
> GraphMask|$0.0767_{\pm 0.0089}$|$0.8244_{\pm 0.0114}$|$0.1402_{\pm 0.0149}$
> IDEA|$\mathbf{0.6881}_{\pm 0.0148}$|$\mathbf{0.9462}_{\pm 0.0087}$|$\mathbf{0.7966}_{\pm 0.0088}$
> *Improvement*|$15.21\\%$|$11.38\\%$|$13.59\\%$
>
> The results above indicate that IDEA can stably surpass the evaluated baselines, demonstrating the effectiveness of IDEA when apply to large real-world graphs from diverse domains.
>
> Regarding the theoretical scalability of IDEA,  we have analyzed its asymptotic time complexity in Appendix E, revealing that IDAE is linear to the node number of the computation graph.

---

> ### Author Response · Authors · 2025-11-19
> **Official Comment by Authors (2/4)**
>
> > **Weakness 2**. Lack of clarity on prototype initialization: The paper does not describe how the prototype embeddings (codewords in $C_S$ and $C_D$) are initialized. This is important for reproducibility and could affect convergence behavior.
>
> > **Question 4**. How are the prototype codebooks $C_S$ and $C_D$ initialized? Is the method sensitive to different initialization schemes?
>
> We apologize for not making this detail clear. In our implementation, the prototype embeddings, i.e., the codewords inside $C_S$ and $C_D$, are initialized by conducting K-means algorithm on the input embeddings of the first batch. This initialization strategy is widely adopted by several codebook-related works [2-4]. During practices, we have attempted uniform distribution initialization and normal distribution initialization, both of which are slightly inferior to the K-means initialization. The explanation performance comparison of the three initialization schemes is presented as follows, which can reveal the insensitivity of IDEA to different schemes.
>
> Initialization|Mutagenicity|Benzene|Alkane|Fluoride|BA-2Motifs|Average
> -|-|-|-|-|-|-
> K-Means|$0.7379_{\pm 0.0084}$|$0.9138_{\pm 0.0002}$|$0.9355_{\pm 0.0030}$|$0.8868_{\pm 0.0018}$|$0.9541_{\pm 0.0107}$|$0.8856_{\pm 0.0047}$
> Uniform|$0.7234_{\pm 0.0067}$|$0.9133_{\pm 0.0003}$|$0.9319_{\pm 0.0104}$|$0.8788_{\pm 0.0040}$|$0.9542_{\pm 0.0024}$|$0.8803_{\pm 0.0036}$
> Normal|$0.7239_{\pm 0.0092}$|$0.9131_{\pm 0.0003}$|$0.9280_{\pm 0.0117}$|$0.8732_{\pm 0.0046}$|$0.9541_{\pm 0.0039}$|$0.8785_{\pm 0.0039}$
>
> > **Weakness 3**. Possible invalid supervision in Equation (5): The second term of Equation (5) uses CrossEntropy between $\hat y_D$ and $\hat y$. However, $\hat y_D$ is generated from the deep quantized representations $q^* _D $, which are OOD with respect to the frozen GNN predictor. If the predictor was trained on original (non-disentangled) representations, its output for $q_D^*$ may not be meaningful.
>
> > **Weakness 4**. Residual OOD inconsistency in optimization: Although the paper claims that aligning in representation space avoids the OOD issue, Equation (5) still uses frozen GNN predictions $\hat y$ as supervision for deep quantized vectors $\hat y_D$, which remain out-of-distribution. This inconsistency needs to be addressed.
>
> > **Question 1**. Regarding Equation (5): Could you justify why minimizing CrossEntropy($\hat y_D$, $\hat y$) is valid given that $\hat y_D$ is OOD with respect to the frozen predictor?
>
> Thanks for your insightful comments. Regarding the CrossEntropy loss in Equation (5), it is a feasible objective to extract explanatory information, as revealed by several previous methods (GNNExplainer, PGExplainer, ReFine, and etc.), despite of its possible invalidity caused by OOD issue. Furthermore, the final explanation provided by IDEA is not dominated by CrossEntropy($\hat y_D$, $\hat y$). Empirically, as shown by Table 9 in Appendix F.3, when we perturb the information disentanglement stage by flipping the GNN prediction $\hat y$ in Equation (5), the explanation performance of IDEA did not decline significantly. Therefore, to some extent, the representation alignment in the prototypical space can mitigate the possible negative impact introduced by CrossEntropy($\hat y_D$, $\hat y$).
>
> > **Weakness 5**. Incomplete related work: The paper largely overlooks existing prototype-based explanation approaches. In addition to Dai & Wang (2025), key missing references include:
> > - ProtGNN (Zhang et al., AAAI 2021)
> > - PAGE (Shin et al., TPAMI 2022)
> > - Prototype-Based Interpretable GNNs (Ragno et al., IEEE TAI 2024)
> > - Towards Prototype-Based Self-Explainable GNN (Dai & Wang, TKDD 2022)
>
> We have carefully read the references and gained a more comprehensive understanding of prototype-based explanation methods. Within the mentioned methods, ProtGNN (AAAI 2021) [5], PAGE (TPAMI 2024) [6], and Towards Prototype-Based Self-Explainable Graph Neural Network (TKDD 2025) [7], focus on self-interpretable graph neural networks, which belong to another research line in GNN explanation methods. Prototype-Based Interpretable Graph Neural Networks (IEEE TAI 2024) [8] belongs to the model-level explanation methods. We have revised the related work section and cited these works to discuss the utilization of prototype in these methods.
>
> By the way, the research paper 'Towards Prototype-Based Self-Explainable GNN (Dai & Wang, TKDD 2022)' you mentioned is the same work to our reference 'Dai & Wang (2025)'. This work is submitted in 2022 and accepted in 2025.

---

> ### Author Response · Authors · 2025-11-19
> **Official Comment by Authors (3/4)**
>
> > **Question 3**. Is it always appropriate to enforce the shallow quantized representation to produce a uniform prediction distribution (first term in Equation (5))?
>
> To the best of our knowledge, the uniform prediction distribution is a reasonable prior to extract the non-explanatory, label-irrelevant substructures, whose effectiveness is validated by previous researches [9-12]. While the uniform prediction distribution might be not always appropriate for all the scenarios, it is a general and feasible prior assumption, when we lack a clear understanding towards the distribution of label-irrelevant substructures [9-12].
>
> > **Question 5**. Can the authors provide qualitative or quantitative analysis of the learned prototypes? E.g., do deep prototypes align with known substructures (e.g., benzene rings)?
>
> Following your insightful comment, in the revised version, we have visualized the assignment distribution, to investigate the relationship between the prototypical embeddings and the human-intelligible substructures. As shown by Figure 13 in Appendix H of the revised version, we present the average probabilistic distributions of class 0 and class 1 over the shallow and deep codebook, respectively. For the real-world dataset Benzene, the distributions of class 0 and class 1 over the shallow codebook are similar. One can see that the codeword 5 with the largest probability may correspond to the most frequent non-explanatory substructure (carbon-chlorine bond). On the deep codebook, the distribution present obviously different patterns. For the deep codebook, the codeword 0 may correspond to the benzene rings which directly decides the labels of class 1, and the codeword 2 may correspond to the carbon-oxygen bond which is common in class 0. For the synthetic dataset BA-2Motifs, the shallow distribution patterns of class 0 and class 1 are also similar. The deep shallow distribution has two peaks, i.e., codeword 1 and codeword 5, which may correspond to the two types of motifs in BA-2Motifs.
>
> To sum up, the similar distribution pattern on shallow codebook and significantly different patterns on deep codebook can indicate that the learned prototypes in codebooks are implicitly related to substructures.
>
> > **Question 6**. Do shallow and deep prototypes exhibit distinguishable structural roles? Can visualizations or clustering be used to support this?
>
> Following your valuable comment, in the revised version, we have presented the t-SNE visualization of learned prototype, to investigate the distinguishable characteristics between shallow and deep prototypes. As shown by Figure 14 and Figure 15 in Appendix H of the revised version, the first row presents the t-SNE visualization of the initial codewords, and the second row presents that of the codewords after optimization, i.e., prototypes. We can notice that in the initial state, the shallow and deep codewords mix together without clear boundary. After optimization, the deep codewords are approximately separable from the shallow ones. The deep codewords prefer to cluster into a mass, while the shallow codewords still distribute dispersedly.
>
> > **Question 7.** Some baseline results seem lower than in original papers. Are all results averaged over multiple runs? Could standard deviations be reported?
>
> We apologize for the missing of standard deviations and we have reported them (Tables 1-3) in the revised version. The results are averaged over ten different random seeds. The fluctuations in the results of the baseline model might be due to the fact that the dataset splits are different from those of their original papers. All the models in our experiment are fairly compared with the same dataset split and random seed set.
>
> > **Question 8**. Hyperparameter tuning varies significantly across datasets (e.g., codebook size $K$). How practical is IDEA in scenarios without access to per-dataset tuning resources?
>
> Empirically, according to the analysis on the codebook size $K$ in Appendix C.2, the recommended value set of $K$ is $\\{16,32,48\\}$, where IDEA achieves the best performance on most of the evaluated datasets.

---

> ### Author Response · Authors · 2025-11-19
> **Official Comment by Authors (4/4)**
>
> > **Question 10**. The paper mentions that node-level tasks can be reformulated as graph-level tasks using computation graphs. Could the authors validate this claim with experiments, or clarify the scope as graph-level only?
>
> Following your valuable comment, we have evaluated IDAE and other baselines on three node classification datasets Cora, PubMed, and Amazon-Computers. The details and experiment results are introduced in our response to Weakness 1.
>
> > **Question 11**. How does IDEA differ fundamentally from prior prototype-based GNN explainers? Is the key contribution the new optimization paradigm (label space → representation space), or the hierarchical disentanglement mechanism?
>
> Compared with previous explainers that are based on prototype, the key contribution of IDEA is the optimization paradigm in the representation space, instead of the label space. Moreover, previous prototype-based GNN explainers mostly focus on self-interpretable GNNs [5,7,8] or model-level explanation methods [6], while we focus on post-hoc instance-level explanation methods.
>
> **Reference**
>
> [1] GraphFramEx: Towards Systematic Evaluation of Explainability Methods for Graph Neural Networks. LoG 2022 and New Frontiers in Graph Learning Workshop of NeurIPS 2022.
>
> [2] VQGraph: Rethinking Graph Representation Space for Bridging GNNs and MLPs. ICLR 2024.
>
> [3] Adapting Large Language Models by Integrating Collaborative Semantics for Recommendation. ICDE 2024.
>
> [4] Unleash LLMs Potential for Sequential Recommendation by Coordinating Dual Dynamic Index Mechanism. WWW 2025.
>
> [5] ProtGNN: Towards Self-Explaining Graph Neural Networks. AAAI 2022.
>
> [6] PAGE: Prototype-Based Model-Level Explanations for Graph Neural Networks. IEEE TPAMI 2024.
>
> [7] Prototype-Based Interpretable Graph Neural Networks. IEEE TAI 2024.
>
> [8] Towards Prototype-Based Self-Explainable Graph Neural Network. ACM TKDD 2025.
>
> [9] Causal Attention for Interpretable and Generalizable Graph Classification. KDD 2022.
>
> [10] Causal-Trivial Attention Graph Neural Network for Fault Diagnosis of Complex Industrial Processes. IEEE Transactions on Industrial Informatics, Volume 20, Issue 2, February 2024.
>
> [11] Enhancing Out-of-distribution Generalization on Graphs via Causal Attention Learning. ACM Transactions on Knowledge Discovery from Data, Volume 18, Issue 5, March 2024.
>
> [12] Graph Out-of-Distribution Generalization via Causal Intervention. WebConf 2024.

---

> ### Comment · Reviewer_dWjn · 2025-11-27
> **Reviewer Response to Rebuttal**
>
> I sincerely thank the authors for their detailed and thoughtful rebuttal, which addresses many of the raised concerns with additional experimental results, clarifications, and theoretical justification.
>
> The authors have provided a comprehensive and convincing rebuttal. Most of the major concerns—including scalability, interpretability of prototypes, related work coverage, and reproducibility—have been addressed either experimentally or through clarification. Minor limitations remain, but these do not significantly detract from the paper's contributions.
>
> I now lean toward recommending acceptance, especially given the novelty of the optimization paradigm and the thoroughness of the additional experiments.

---

> > ### Author Response · Authors · 2025-11-28
> >
> > Dear Reviewer dWjn,
> >
> > We are truly grateful for your thoughtful reevaluation and the increase in your score. Your constructive feedback and suggestions are insightful in improving our work.
> >
> > Best regards,
> > Authors of paper 858

---

### Author Response · Authors · 2025-11-25

Dear Reviewers,

We sincerely appreciate your valuable comments and thoughtful suggestions, which have greatly helped us refine our work. Following your advice, we have provided additional clarifications and further experiments. We hope these efforts address your concerns effectively.

Since half of the discussion period has elapsed, we sincerely look forward to your reevaluation and would be deeply grateful if you could consider a higher rating. Thank you very much for your time and consideration!

Best regards,

Authors of paper 858

---

### Comment · Area_Chair_JA2B · 2025-11-26
**check author's response**

Dear reviewers,

as the authors/reviewers discussion is ending soon, please check the authors' rebuttal and update your reviews/scores accordingly.  Thank you for your service to ICLR.


Best,
AC

---

### Meta-Review · Area_Chair_Az3g · 2026-01-07

**Summary:**

The paper proposes a novel post-hoc instance-level GNN explainer. Instead of identifying a compact subgraph in original graph space, the authors designed HGTokenizer to tokenize and disentangle the graph  into explanatory and non-explanatory substructures, and then optimize the explainer. Experimental results demonstrate consistent performance improvements across multiple datasets.

The reviewers raised concerns spanning dataset coverage, HGTokenizer design, theoretical justification, loss formulation, prototype-related related work, supervision and disentanglement clarity, and Fidelity-based evaluation metrics. In response, the authors added larger real-world node classification datasets, incorporated Fidelity-based evaluations, clarified the Wasserstein matrix and supervision paths, explored a HGTokenizer variant to justify the design, provided detailed implementation descriptions, differentiated their approach from model-level prototype methods, and supplemented all experimental results with standard deviations. Overall, the authors have addressed the majority of reviewers’ concerns in a thorough and satisfactory manner.

Recommendation:  accept as a poster

**Reviewer Concerns:**

Reviewer dWjn raised concerns about dataset coverage, initialization details, potential confusion in the loss design, missing prototype-based related work, and hyperparameter analysis. In response, the authors added three larger real-world node classification datasets and adopted Fidelity-based metrics for evaluation, provided detailed implementation descriptions, and clarified the distinctions between their method and existing model-level prototype-based approaches.

Reviewer dsYU questioned the HGTokenizer design, the theoretical justification, and other evaluation metrics such as fidelity. The authors addressed these concerns by trying a variant of HGTokenizer to justify the design choices and use Fidelity-based evaluations across diverse datasets.

Reviewer GAVp expressed concerns about the Wasserstein cost matrix, the supervision path, and potential leakage in the disentanglement, and the absence of standard deviations. The authors clarified these methodological details and supplemented all experimental tables with standard deviations.

Reviewer u3DN raised questions regarding the design of the HGTokenizer and the potential extension of the framework to model-level explanations. The authors provided clarifications.

Overall, the authors have addressed the majority of the reviewers’ concerns in a thorough and satisfactory manner.

**Reviewer Scores:**

Reviewer dWjn raised the score and expressed a lean toward recommending acceptance. The other reviewers appear to have maintained their original scores.

---

### Decision · Program_Chairs · 2026-01-26

Accept (Poster)